# Germline loss in *C. elegans* enhances longevity by disrupting adhesion between niche and stem cells

Meng Liu[1,2], Jiehui Chen (ID)[1], Guizhong Cui (ID)[1,3], Yumin Dai[1], Mengjiao Song[1], Chunyu Zhou[4,5], Qingyuan Hu[1,2], Qingxia Chen[6], Hongwei Wang[1,2], Wanli Chen[1,2], Jingdong Jackie Han (ID)[5], Guangdun Peng[2,3,7], Naihe Jing (ID)[1,2,3,7] & Yidong Shen (ID)[1,2]✉

## Abstract

**Ageing and fertility are intertwined. Germline loss extends the lifespan in various organisms, termed gonadal longevity. However, the original longevity signal from the somatic gonad remains poorly understood. Here, we focused on the interaction between germline stem cells (GSCs) and their niche, the distal tip cells (DTCs), to explore the barely known longevity signal from the somatic gonad in *C. elegans*. We found that removing germline disrupts the cell adhesions between GSC and DTC, causing a significant transcriptomic change in DTC through *hmp-2*/β-catenin and two GATA transcription factors, *elt-3* and *pqm-1* in this niche cell. Inhibiting *elt-3* and *pqm-1* in DTC suppresses gonadal longevity. Moreover, we further identified the TGF-β ligand, *tig-2*, as the cytokine from DTC upon the loss of germline, which evokes the downstream gonadal longevity signalling throughout the body. Our findings thus reveal the source of the longevity signalling in response to germline removal, highlighting the stem cell niche as a critical signalling hub in ageing.**

**Keywords** Gonadal Longevity; Niche; Cell Adhesion; *C. elegans*
**Subject Categories** Development; Signal Transduction; Stem Cells & Regenerative Medicine

## Introduction

Reproduction and ageing tightly interact with each other. It has been shown in various organisms that the absence of germline significantly extends lifespan (Antebi, 2013; Flatt et al, 2008; Hsin and Kenyon, 1999; Kenyon, 2010; Min et al, 2012). Studies in the nematode *Caenorhabditis elegans* indicate that the somatic gonad generates an unknown signal to trigger a complex signalling network in other tissues to promote longevity when the germline is removed (Antebi, 2013; Hsin and Kenyon, 1999). Downstream of the somatic gonad-derived signal lies a complex genetic network (Antebi, 2013). For example, *daf-16*/FOXO controls gonadal longevity (Berman and Kenyon, 2006; Lin et al, 2001). The biosynthesis of dafachronic acids (DAs) and the subsequent activation of the nuclear hormone receptor DAF-12/FXR is another critical pathway driving gonadal longevity (Gerisch et al, 2007; Shen et al, 2012; Wollam et al, 2011). Intriguingly, the gonadal longevity signalling shares components with developmental timing machinery. In particular, the DA synthesis and DAF-12 activation are initiated at the end of germline development (Shen et al, 2012), implying that gonadal longevity could be from a checkpoint for germline integrity.

Despite the extensive understanding of the molecules controlling ageing upon germline ablation, the longevity signal from the somatic gonad remains poorly understood. Within the germline, the somatic gonad constitutes the niche of germ cells and regulates their development (Xie, 2008). It is the germline stem cells (GSCs) but not the oocytes or sperms that influence ageing (Arantes-Oliveira et al, 2002). Therefore, we hypothesize that the gonadal longevity signal originates from the somatic gonadal cells neighbouring GSCs because these cells have intensive interactions with GSCs as their niche and should be the first to sense their absence (Xie, 2008).

In this study, we found that removing worm germline disrupts the cell adhesions between GSC and its niche, the distal tip cell (DTC), causing a significant transcriptomic change in DTC through the translocation of two GATA transcription factors, *elt-3* and *pqm-1*, and the translocation of *hmp-2*/β-catenin. This, in turn, extends the lifespan of worms. Moreover, we further identified the TGF-β ligand, *tig-2*, as the cytokine from DTC upon germline ablation, which evokes the downstream longevity pathways throughout the body. Our findings thus reveal the origin of the longevity signalling in response to germline ablation, underscoring the interaction of stem cells and their niche in metazoan ageing.

[1]State Key Laboratory of Cell Biology, Shanghai Institute of Biochemistry and Cell Biology, Center for Excellence in Molecular Cell Science, Chinese Academy of Sciences, 200031 Shanghai, China. [2]University of Chinese Academy of Sciences, 100049 Beijing, China. [3]Guangzhou Laboratory, 510005 Guangzhou, China. [4]CAS Key Laboratory of Computational Biology, Shanghai Institute of Nutrition and Health, Shanghai Institutes for Biological Sciences, Chinese Academy of Sciences, 200031 Shanghai, China. [5]Peking-Tsinghua Center for Life Sciences, Academy for Advanced Interdisciplinary Studies, Center for Quantitative Biology (CQB), Peking University, 102213 Beijing, China. [6]Ministry of Education-Shanghai Key Laboratory of Children's Environmental Health, Institute of Early Life Health, Xinhua Hospital, Shanghai Jiao Tong University School of Medicine, 200092 Shanghai, China. [7]Center for Cell Lineage and Development, CAS Key Laboratory of Regenerative Biology, Guangdong Provincial Key Laboratory of Stem Cell and Regenerative Medicine, GIBH-HKU Guangdong-Hong Kong Stem Cell and Regenerative Medicine Research Centre, Guangzhou Institutes of Biomedicine and Health, Chinese Academy of Sciences, 510530 Guangzhou, China. ✉E-mail: yidong.shen@sibcb.ac.cn

# Results

## The loss of germline induces transcriptomic changes in the GSC niche

DTC at the end of the distal gonad in worms forms the niche encasing GSCs (Kimble and White, 1981). We isolated DTCs from both wild-type (WT) worms and the germlineless *glp-1* mutants at day 1 of adulthood for RNA-Seq (Arantes-Oliveira et al, 2002; Peng et al, 2019) (Fig. EV1A). RT-qPCR of tissue-specific genes showed that the isolated DTCs were of little contamination from other tissues (Fig. EV1B,C). A comparison with genes specifically detected in five other worm tissues confirmed the purity of isolated DTCs (Wang et al, 2022) (Fig. EV1D). Reads quality, the number of detected genes, and RT-qPCR of representative differentially expressed genes (DEGs) further confirmed the quality of the DTC-specific RNA-Seq datasets (Fig. EV1E–G).

Principal component analysis and hierarchical clustering of the DTC-specific RNA-Seq datasets indicated that the transcriptome in DTCs underwent remarkable changes when the germline was removed by mutating *glp-1* (Fig. 1A,B, Dataset EV1). Among the 20,486 genes identified in DTCs, the loss of germline induced significant upregulation of 1277 (6.2%) and downregulation of 966 (4.7%) genes (Fig. 1C). Notably, the DEGs in DTCs were overlapping but different from those in the whole worm (Nakamura et al, 2016) (Fig. 1C). Gene set enrichment analysis (GSEA) by WormCat showed that germline removal altered similar biological pathways in DTC and in the worm (Fig. 1D and Dataset EV2). Some of these pathways, such as 'stress response', 'metabolism' and 'proteolysis', are known to regulate ageing (Lopez-Otin et al, 2013). Nevertheless, the upregulated genes in DTC were also enriched in a few specific pathways, including 'signalling' (Fig. 1D and Dataset EV2), implying that the germline removal could trigger unique changes in DTC other than general longevity phenotypes.

By analysing the consensus sites in the promoter regions of DEGs with HOMER (Heinz et al, 2010), we found that the GATA transcription factors (TFs), ELT-3 and PQM-1, could regulate 46.9% and 41.9% upregulated genes, respectively (Fig. 1E,F). Both TFs are reported to regulate ageing (Budovskaya et al, 2008; Tepper et al, 2013). Interestingly, the binding motifs of the two TFs are highly similar, with a difference of only one nucleotide (Fig. 1E). Consequently, their targets are largely overlapping, constituting 51.6% of upregulated genes in DTC upon germline removal (Fig. 1F). The expression of ELT-3::mCherry and PQM-1::mCherry in DTC showed no difference between WT worms and *glp-1* mutants (Fig. EV2), suggesting that they could be regulated through their interactors.

## PQM-1 and ELT-3 in DTC induce gonadal longevity

Gonadal longevity is triggered at the fourth larval stage (L4), concurring with the start of large-scale germ cell proliferation (Hubbard and Greenstein, 2005; Shen et al, 2012). Therefore, DTC is likely to emit longevity signals in response to the absence of adjacent GSCs at L4. Following this speculation, we performed DTC-specific RNAi against *elt-3* or *pqm-1* from L3 to examine whether the two TFs in DTC control gonadal longevity (Linden et al, 2017) (Fig. EV3). Indeed, the longevity induced by

the laser-ablation of germline was also blocked by *elt-3* or *pqm-1* RNAi in DTC (Fig. 2A,B). Similarly, DTC-specific inhibition of *elt-3* or *pqm-1* abolished the extended lifespan in the *glp-1* mutants (Fig. 2C). Mutating *elt-3* or *pqm-1* also suppressed the longevity of *glp-1* mutants (Fig. 2D). Therefore, *elt-3* or *pqm-1* in DTC promotes longevity upon the loss of germline.

DA/*daf-12* pathway and *daf-16* signalling are essential in gonadal longevity (Antebi, 2013). Upon the loss of germline, the critical enzyme in DA biosynthesis, *daf-36*/Rieske-like oxygenase, and *daf-16* target genes are upregulated to drive longevity (Berman and Kenyon, 2006; Gerisch et al, 2007; Hsin and Kenyon, 1999; Shen et al, 2012; Wang et al, 2008; Wollam et al, 2011). To examine whether DTC controls DA/*daf-12* and *daf-16* signalling, we examined the expression of *daf-36* and a group of *daf-16* targets (i.e., *lipl-4*, *dod-8*, and *sod-3*) upon DTC-specific RNAi against *elt-3* or *pqm-1*. In line with their regulation on lifespan (Fig. 2B,C), suppressing *elt-3* or *pqm-1* in DTC significantly inhibited the upregulation of *daf-36* and *daf-16* targets in *glp-1* mutants (Fig. 2E), indicating that DTC induces both DA/*daf-12* and *daf-16* signalling in response to germline loss. Mutating *elt-3* or *pqm-1* also suppressed the induction of *daf-36* and *daf-16* targets in *glp-1* mutants (Fig. 2F), confirming their requirements in gonadal longevity signalling.

## DTC perceives the absence of GSCs through cell adhesions

DTC is a claw-like cell enwrapping GSCs and other germ cells with its soma and numerous dendritic protrusions. Worm GSCs physically contact DTC through E-cadherin-based cell adhesions to retain their stem cell properties (Ferraro et al, 2010; Gordon et al, 2019) (Fig. 3A). *hmr-1* is the worm ortholog of E-cadherin (Costa et al, 1998). As reported (Gordon et al, 2019), we observed in DTC that endogenously GFP-tagged HMR-1 formed cell adhesions on the cell membrane facing GSCs (Fig. 3B, Movie EV1 and EV2). Moreover, HMR-1::GFP-labelled cell adhesions were also present on the dendritic structures of DTC (Fig. 3B and Movie EV2).

In *glp-1* mutants deficient of GSCs, DTC underwent a remarkable morphological change as reported (Linden et al, 2017), with most of its dendritic structures diminishing (Fig. 3B, Movie EV1 and EV2), suggesting a dramatic change in its cell adhesions. We next focused on the relatively unchanged cell body of DTC and found the enrichment of HMR-1::GFP-labelled adhesions was reduced remarkably (Fig. 3B, Movie EV1 and EV2), whereas a non-relevant membrane bound mCherry-PH protein remained unchanged (Fig. EV4A). Considering the loss of DTC dendritic structures and the cell adhesions on them (Fig. 3B), the actual decrease of DTC-GSC adhesions could be more severe than our observation in DTC cell body (Fig. 3B). Therefore, DTC could sense the absence of GSCs via the loss of the in-between cell adhesions.

Disrupting the DTC-GSC adhesions via DTC-specific *hmr-1* RNAi in WT worms from L3 extended lifespan (Figs. 3C and EV4B) and upregulated *daf-36* and the target genes of *daf-16* (Fig. 3D), supporting that the DTC-GSC adhesions regulates gonadal longevity. The DTC-GSC adhesions were already disrupted upon germline removal (Fig. 3B). We then knocked down *hmr-1* in the DTC of *glp-1* mutants to confirm whether it controls gonadal longevity through the DTC-GSC adhesions. If so, the DTC-specific

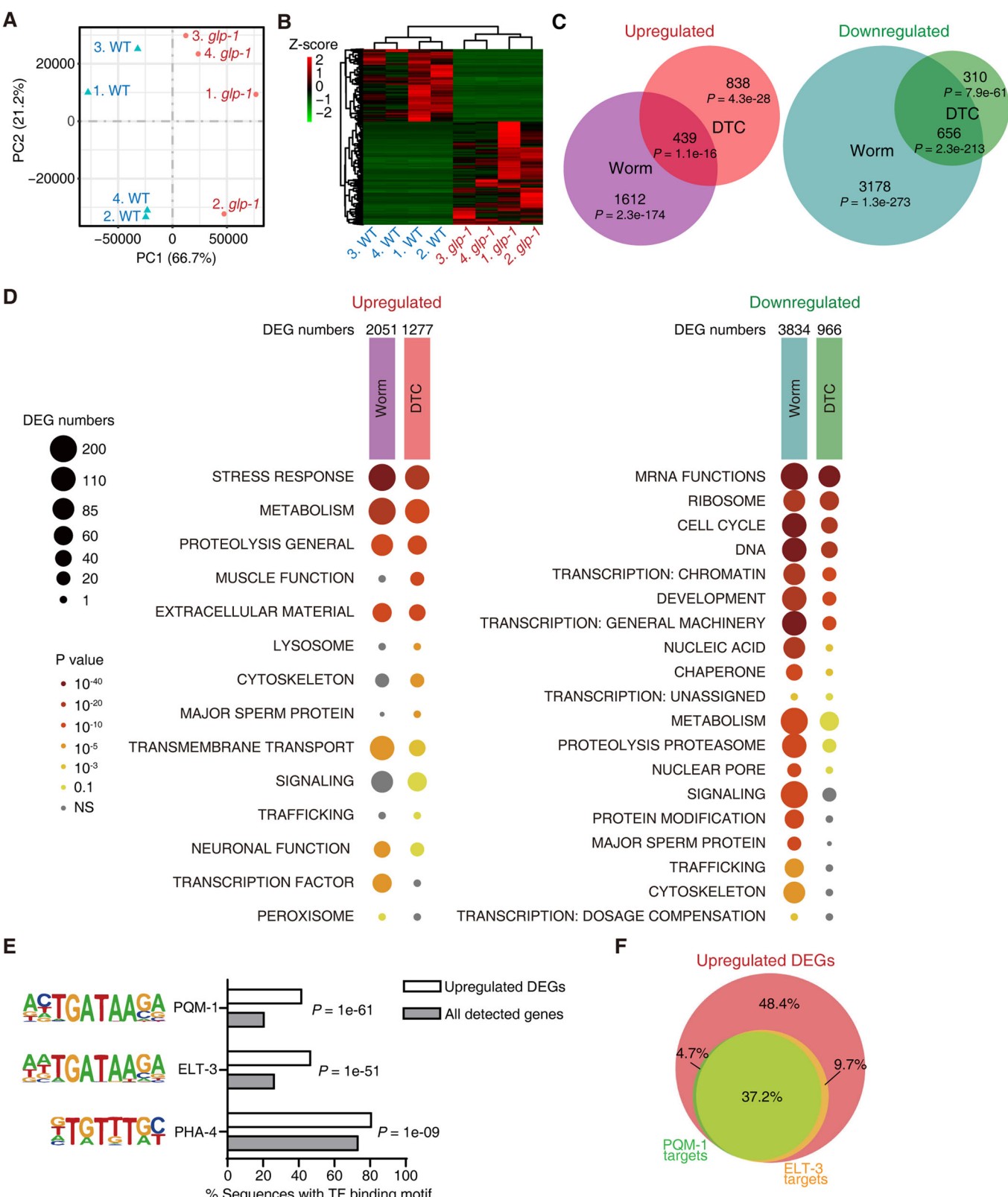

**Figure 1. The loss of germline induces a substantial transcriptomic change in DTCs.**

(A) Principal component analysis (PCA) of the mRNA profiling in the distal tip cells (DTCs) of indicated strains at day 1 of adulthood. Component 1 accounts for 66.7% of the variation. *glp-1* is a mutant without germline. (B) A heatmap depicting the differential expression of all responsive genes in the DTCs of indicated strains. (C) Venn diagrams comparing the differentially expressed genes (DEGs) in the whole worm and DTC upon the mutation of *glp-1*. Hypergeometric probability test. (D) Gene set enrichment analysis of the DEGs in DTC upon the loss of germline. Analysis was performed using WormCat. Category 1 is shown. See Dataset EV2 for the information of Categories 2 and 3. Fisher's exact test. (E) Top three transcription factors (TFs) predicted to drive the upregulated DEGs in *glp-1* DTCs by HOMER. TFs were identified and ranked by the enrichment of their binding motifs in the gene promoters. All detected genes in DTC serve as background. Hypergeometric probability test. (F) A comparison of predicted targets of ELT-3 and PQM-1 in the upregulated genes in the DTC of *glp-1* mutants.

RNAi against *hmr-1* should have no corresponding phenotypes. Indeed, inhibiting *hmr-1* in the DTC of *glp-1* mutants did not affect lifespan or the expression of *daf-36* or *daf-16* targets (Figs. 3D and EV4C).

To further examine the role of the DTC-GSC adhesions in gonadal longevity, we examined its epistasis with *elt-3* and *pqm-1* in DTC. If suppressing *elt-3* or *pqm-1* in DTC blocks the *hmr-1* RNAi-induced phenotypes, *hmr-1* (i.e., the DTC-GSC adhesions) should function upstream of these two TFs. Double RNAi against *hmr-1* and *elt-3*, or *hmr-1* and *pqm-1* in DTC abolished the extended lifespan by DTC-specific *hmr-1* RNAi (Fig. 3E), while efficiently disrupting DTC-GSC adhesions (Fig. EV4D). The increased expression of *daf-36* and *daf-16* targets upon the DTC-specific suppression of *hmr-1* was similarly inhibited in the worms treated by double RNAi (Fig. 3F). Therefore, the loss of GSCs activates the longevity signal in DTC through reducing the DTC-GSC adhesions.

## β-catenin activates GATA TFs in DTC upon the loss of DTC-GSC adhesions

β-catenin binds to E-cadherin and is translocated from the cell membrane into the nucleus upon the disruption of cell adhesions (Orsulic et al, 1999). Besides, β-catenin is reported to interact with GATA TF to regulate its activity (Iyer et al, 2018). Therefore, the loss of the DTC-GSC adhesions upon germline removal could trigger *elt-3* and *pqm-1* through β-catenin in DTC.

HMP-2, orthologous to β-catenin, interacts with HMR-1/E-cadherin and is localised at the DTC-GSC adhesion (Costa et al, 1998; Gordon et al, 2019). Like HMR-1::GFP, the HMP-2 complex, labelled by endogenously tagged GFP::HMP-2, decreased on the DTC membrane facing GSCs in *glp-1* mutants (Fig. 4A). DTC-specific RNAi against *hmr-1* consistently suppressed GFP::HMP-2 on DTC membrane facing GSCs (Fig. 4B). Because the fluorescence of the endogenously GFP-tagged HMP-2 is too weak to examine its nuclear localization (Fig. 4A,B), we next pursued its nuclear translocation by an extrachromosomal array and a more potent fluorescent tag, mGreenLantern (mGL) (Campbell et al, 2020). As we speculated, the nuclear localization of mGL-tagged HMP-2 in DTC was increased in *glp-1* mutants, compared to that in WT worms (Fig. EV4E). Moreover, GFP::HMP-2 was co-immunoprecipitated with ELT-3::FLAG and PQM-1::FLAG (Fig. EV4F,G), suggesting HMP-2 could interact with the two GATA TFs in DTC. Therefore, HMP-2 translocates into DTC nucleus and activates the two GATA TFs upon the loss of germline.

We further examined the lifespan of WT worms and *glp-1* mutants upon DTC-specific RNAi of *hmp-2*. Inhibiting *hmp-2* in DTC suppressed the longevity of *glp-1* mutants as expected (Fig. 4C). The upregulation of *daf-36* and *daf-16* targets in *glp-1*

mutants were accordingly inhibited by DTC-specific RNAi against *hmp-2* (Fig. 4D). These results then indicate that *hmp-2* functions in the signalling from the DTC-GSC adhesions to GATA TFs in DTC.

## A TGF-β ligand from DTC regulates longevity upon the loss of germline

There are only two DTCs in *C. elegans*, which has around one thousand somatic cells. Therefore, DTCs must signal to other tissues to modulate worm longevity. The signalling molecules will likely be increased in DTCs when the germline is removed. Following this speculation, we selected the secreted proteins upregulated by at least two-fold in the DTC of *glp-1* mutants, screened for the regulator of gonadal longevity by DTC-specific RNAi in *glp-1* mutants, and found a TGF-β ligand, *tig-2* (Gumienny and Savage-Dunn, 2013) (Dataset EV1). TGF-β signalling is critical in ageing (Luo et al, 2010; Luo et al, 2009; Shaw et al, 2007). Among the four TGF-β ligand genes detected in DTC, *tig-2* shows the highest expression (Fig. EV5A). Its promoter harbours ELT-3 and PQM-1 binding sites (Fig. EV5B). The two GATA TFs co-immunoprecipitated with *tig-2* promoter and activated a luciferase reporter driven by it in HEK293T cells (Figs. 5A and EV5C), showing that the two GATA TFs can directly interact with the promoter of *tig-2*. TIG-2::mCherry in the DTC of *glp-1* mutants is increased, compared with that of WT worms (Fig. EV5D). Moreover, RT-qPCR on isolated DTCs showed that *tig-2* was upregulated in the DTC of *glp-1* mutants, as the RNA-Seq data suggested (Figs. 5B and EV5E, and Dataset EV1). DTC-specific RNAi against *elt-3* or *pqm-1* suppressed the increase of *tig-2* in DTC (Figs. 5B and EV5E), confirming that *tig-2* is induced by the two GATA TFs upon the removal of germline in vivo.

Inhibiting *tig-2* specifically in DTC from L3 suppressed the upregulated *daf-36* and *daf-16* targets in *glp-1* mutants (Fig. 5C). Furthermore, its RNAi in DTC abrogated the extended lifespan in *glp-1* mutants (Fig. 5D). These results, therefore, indicate that *tig-2* is required for the DTC-derived longevity signal upon the loss of germline. We further constructed a strain overexpressing *tig-2* by a DTC-specific promoter (Fig. EV5F,G). Besides DTC, the over-expressed TIG-2 was also observed in coelomocytes, the scavenger cells continuously taking in pseudocoelom fluids (Fig. EV5F), indicating that TIG-2 is secreted out from DTC. As expected, DTC-specific upregulation of *tig-2* extended lifespan in WT worms (Fig. 5E,F), showing that *tig-2* from DTC is sufficient to promote longevity. To confirm whether *tig-2* functions as a secreted factor to trigger gonadal longevity, we artificially overexpressed TIG-2::mCherry in the intestine, a tissue with little *tig-2* expression

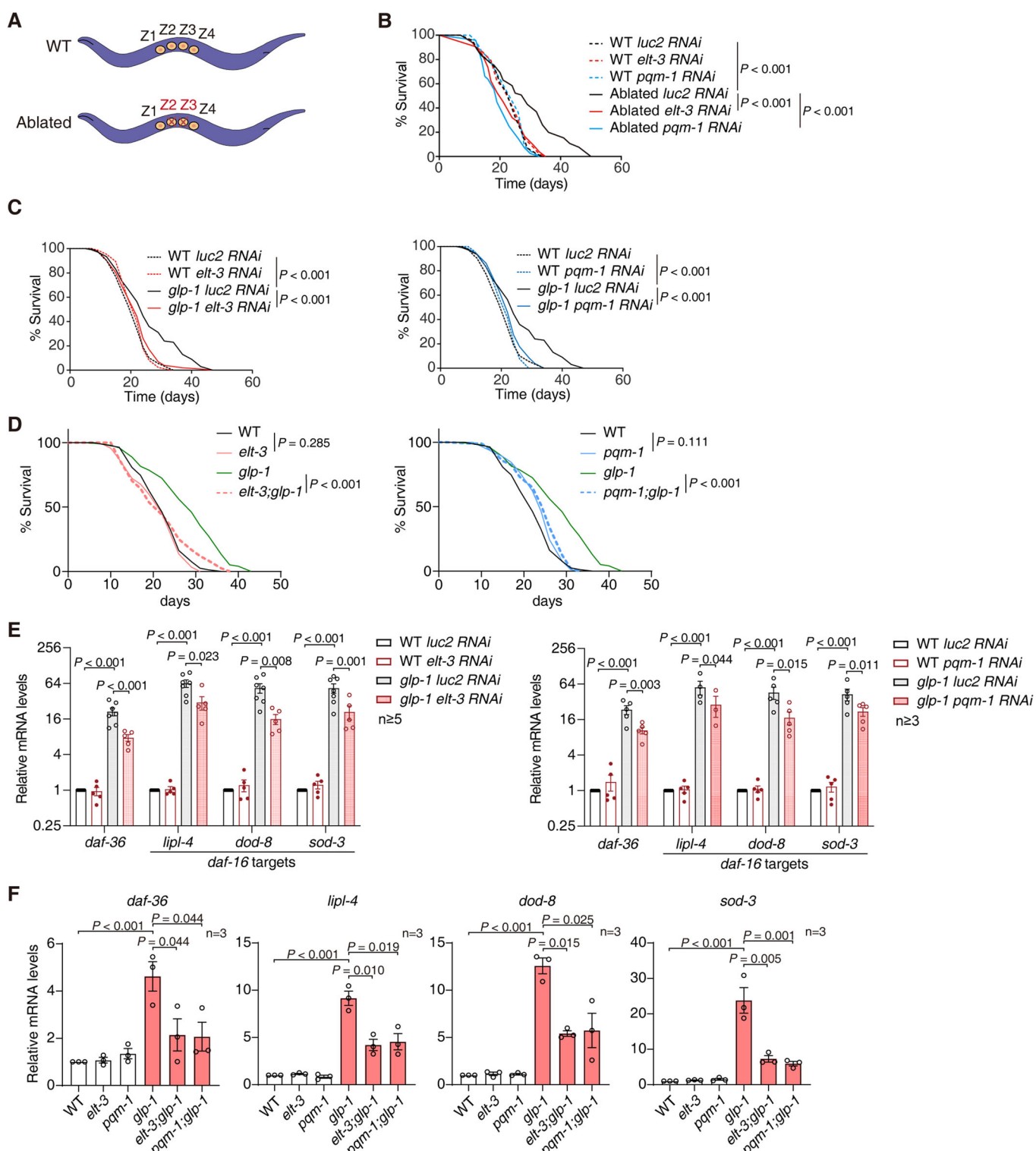

(Wang et al, 2022), and found worms lifespan was indeed increased upon the intestinal overexpression of *tig-2* (Fig. EV5H).

To further examine the position of *tig-2* in gonadal longevity signalling, we first examined its interaction with *hmr-1*, *hmp-2*, and the two GATA TFs. DTC-specific RNAi against these genes had no effect on the extended lifespan upon the overexpression of *tig-2* in

DTC (Figs. 5E and EV5I), confirming that *tig-2* functions downstream of the *hmr-1*-*hmp-2*-GATA TF axis. Moreover, mutating *daf-16* or *daf-36* abrogated the longevity induced by DTC-specific upregulation of *tig-2* (Fig. 5F), indicating that *tig-2* works upstream of these known gonadal longevity signalling in somatic tissues.

**Figure 2.  Two GATA transcription factors in DTC regulate gonadal longevity.**

(**A**) A depiction of the laser ablation to remove the worm germline. Z1 and Z4 cells develop into the somatic gonad, whereas Z2 and Z3 into the germline. (**B**) DTC-specific RNAi against *elt-3* or *pqm-1* abrogates the extended lifespan upon the laser ablation of the germline. Laser ablation of the germline extends the median lifespan by 16.0% upon DTC-specific RNAi against *luc2*, whereas has no effect when *elt-3* or *pqm-1* is inhibited. (**C**) DTC-specific RNAi against the GATA TF *elt-3* (left) or *pqm-1* (right) suppresses the extended lifespan of *glp-1* mutants (12.7% extension in median lifespan). (**D**) Survival curves of indicated strains. A representative biological replicate is shown. (**E**) The mRNA levels of *daf-36*, and *daf-16* target genes in WT or *glp-1* mutants with DTC-specific RNAi against indicated genes at day 1 of adulthood. Error bar: SEM. At least 3 biological replicates were examined. (**F**) RT-qPCR of *daf-36* and three *daf-16* target genes (*lipl-4*, *dod-8*, and *sod-3*) in the indicated strains. Error bar: SEM. 3 biological replicates were examined. The firefly luciferase gene, *luc2*, serves as the negative control in RNAi assays. Mantel-Cox test in (**B**) to (**D**), unpaired *t*-test in (**E**), one-way ANOVA with Tukey's test in (**F**). A representative biological replicate is shown for lifespan analyses in (**B**) and (**C**). See source data for other biological replicates of (**B**–**D**) and detailed statistics. Source data are available online for this figure.

## Discussion

In this study, we show that DTC, the niche of GSCs, is the source of gonadal longevity signalling. The loss of GSCs disrupts the cell adhesions between GSC and DTC, leading to the release of HMP-2/β-catenin in DTC from the cell membrane and the transcriptomic changes in DTC though GATA transcription factors ELT-3 and PQM-1. The TGF-β ligand, TIG-2, is thereby upregulated in and secreted from DTC, activating downstream signalling (e.g., DA signalling) in other somatic tissues to promote longevity. Conversely, in the DTC of an intact germline, HMP-2/β-catenin is retained at the DTC-GSC adhesions, ELT-3 and PQM-1 are not altered, and the expression and secretion of TIG-2 are low. Thus, gonadal longevity signalling is off, resulting in a normal lifespan (Fig. 5G).

Ageing and fertility are closely linked. The loss of germline is proposed to induce an unknown longevity signal in the somatic gonad (Kenyon, 2010). The somatic gonad not only encases the germline but also serves as its microenvironment (Lehmann, 2012). Hence, we examined the germline and the somatic gonad from the perspective of germ cells and their niche cells. Among all the germ cells, GSCs are essential to gonadal longevity (Arantes-Oliveira et al, 2002). DTC is the primary niche of GSC in *C. elegans* (Lehmann, 2012). As the interaction of stem cells and their niche is mutual and critical to each other (Chacon-Martinez et al, 2018; Lehmann, 2012), removing GSC could thus change DTC and induce the gonadal longevity signal.

There are only two DTCs in a worm. Previous transcriptomic studies on the whole worm could thus miss the somatic gonad-derived signal (Nakamura et al, 2016; Steinbaugh et al, 2015). Therefore, we profiled gene expression in isolated DTCs to test our hypothesis. Indeed, DTC undergoes a remarkable transcriptomic change when GSCs are removed (Fig. 1). Compared with the whole worm (Nakamura et al, 2016), DTC shows common and characteristic gene expression upon the loss of germline. The transcriptomic change in DTC could first manifest the tissue-specific longevity programs, as shown in worm and mouse tissues (Tabula Muris, 2020; Wang et al, 2022; Wang et al, 2024). Moreover, it also comprises the longevity signal from the somatic gonad, because suppressing *elt-3* and *pqm-1*, two GATA TFs predicted to upregulate gene expression in DTC, substantially inhibits gonadal longevity and downstream gonadal longevity signalling (Antebi, 2013) (Figs. 1 and 2). DTC-specific RNAi against either GATA TFs does not reduce DA/*daf-12* and *daf-16* signalling to the WT level, implying a redundancy of these two TFs. Nevertheless, such a reduction is enough to reduce the extended lifespan in germlineless worms to WT level, indicating that gonadal longevity does not require the full activation of the complex gonadal longevity pathways but rather a threshold of their collaborative activity. Similarly, *mml-1/mxl-2* regulates gonadal longevity but does not affect the *pha-4/FOXA* expression or DAF-16/FOXO nuclear localization (Lapierre et al, 2011; Lin et al, 2001; Nakamura et al, 2016). In addition to DTC, sheath cells in the somatic gonad interact with GSCs, as well (Killian and Hubbard, 2005; Starich et al, 2014). It will be interesting to test whether sheath cells also signal longevity upon the loss of GSCs.

E-cadherin-based cell adhesions are critical in stem cell-niche interaction. The intercellular signalling by E-cadherin is bidirectional, providing a perfect means for stem cells and niche cells to sense each other (Chacon-Martinez et al, 2018; Chen et al, 2013). β-catenin released from cell adhesions is also a pivotal signalling molecule driving various cellular responses (Valenta et al, 2012). Indeed, removing GSCs remarkably reduces the DTC-GSC adhesions. We further found that DTC responds to the absence of GSCs via the E-cadherin-β-catenin axis (Figs. 3, 4, and EV4). DTC-specific RNAi against *hmr-1*/E-cadherin does not extend lifespan or activate downstream gonadal longevity signalling as significant as the loss of GSCs. This could either be due to the insufficient RNAi efficiency, or suggest other GSC-niche communication (e.g., WNT) is involved (Xie, 2008). Similar to a previously report (Iyer et al, 2018), we found that HMP-2/β-catenin could interact with the two GATA TFs, which drives the transcriptomic changes in DTC (Figs. 1 and EV4), suggesting that HMP-2 might directly regulate the two GATA TFs. The detailed mechanism underlying the HMP-2-mediated regulation on the two GATA TFs upon germline removal will be an interesting issue in future studies.

We find that *tig-2*, a ligand in TGF-β signalling, activates downstream gonadal longevity pathways (e.g., DA signalling) from DTC (Fig. 5). Of note, TGF-β also activates DA signalling to promote reproduction at the dauer checkpoint (Fielenbach and Antebi, 2008). Moreover, TGF-β signalling, like many other signals from the stem cell niche, is crucial for stem cells maintainence, promotes GSC mitosis, and sustain germline quality (Chacon-Martinez et al, 2018; Dalfo et al, 2012; Liu et al, 2018; Luo et al, 2010). Various cues for the stemness and proliferation of stem cells are linked with longevity (Brunet et al, 2022). Therefore, gonadal longevity could result from a quality check for GSC by somatic gonad. When the germline fails the quality check upon the loss of GSCs, DTC signals to the absent GSCs, in an attempt to rescue reproduction while activating longevity signalling in other tissues.

Consistent with this hypothesis, DA signalling is not activated upon germline removal until the fourth larval stage (L4), when the germline is supposed to be mature and ready for quality assessment before reproduction (Shen et al, 2012). Similarly, disrupting DTC

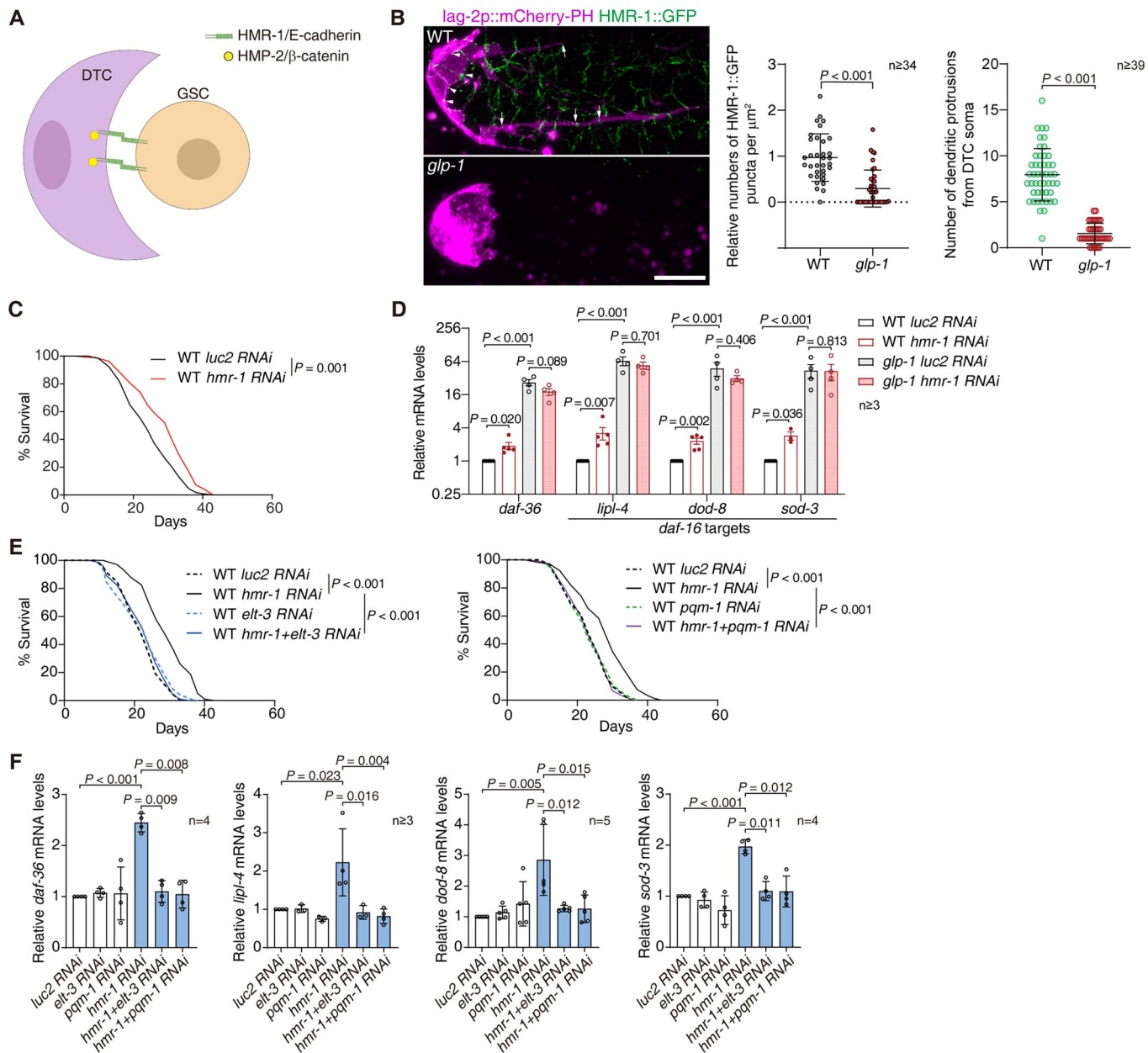

**Figure 3. The loss of cell adhesions between DTC and GSC induces gonadal longevity.**

(A) A diagram showing the E-cadherin-based cell adhesions between DTC and germline stem cells (GSCs). (B) Germline removal reduces the number of the DTC-GSC adhesions and the dendritic protrusions of DTC. The worm ortholog of E-cadherin, HMR-1, was endogenously tagged by GFP. lag-2p::mCherry-PH labels DTC membrane. Note the obvious reduction of HMR-1::GFP labelled DTC-GSC adhesions at both DTC soma (arrowheads) and dendritic protrusions (arrows). DTC morphology undergoes significant change with a remarkable reduction in dendritic protrusions in glp-1 mutants. Representative Z-stacking of confocal microscopic images are shown. Scale bar: 10 μm. Also see Movie EV1 and EV2. The density of HMR-1::GFP puncta on the DTC soma surface towards GSC (arrows) and the number of dendritic protrusions from DTC soma were measured. At least 34 biological replicates (DTCs) were examined. Error bars: SD. (C) DTC-specific RNAi against hmr-1 extends lifespan (7.8% extension in median lifespan). (D) Inhibiting hmr-1 in DTC upregulates the expression of indicated genes in WT worms but not in glp-1 mutants. Error bars: SEM. At least 3 biological replicates were examined. (E, F) Survival curves (E) and the expression of indicated genes (F) in WT worms subjected to indicated DTC-specific RNAi treatments. Note that elt-3 or pqm-1 in DTC is required for the extended lifespan (24.7% extension in median lifespan) and increased expression of daf-36 and daf-16 targets induced by DTC-specific RNAi against hmr-1. At least 3 biological replicates were examined in (F). Error bars: SD. The firefly luciferase gene, luc2, serves as the negative control in RNAi assays. Unpaired t-test in (B) and (D), Mantel-Cox test in (C) and (E), paired t-test in (F). A representative biological replicate is shown for lifespan analyses in (C) and (E). See source data for other biological replicates of (C) and (E) and detailed statistics. Source data are available online for this figure.

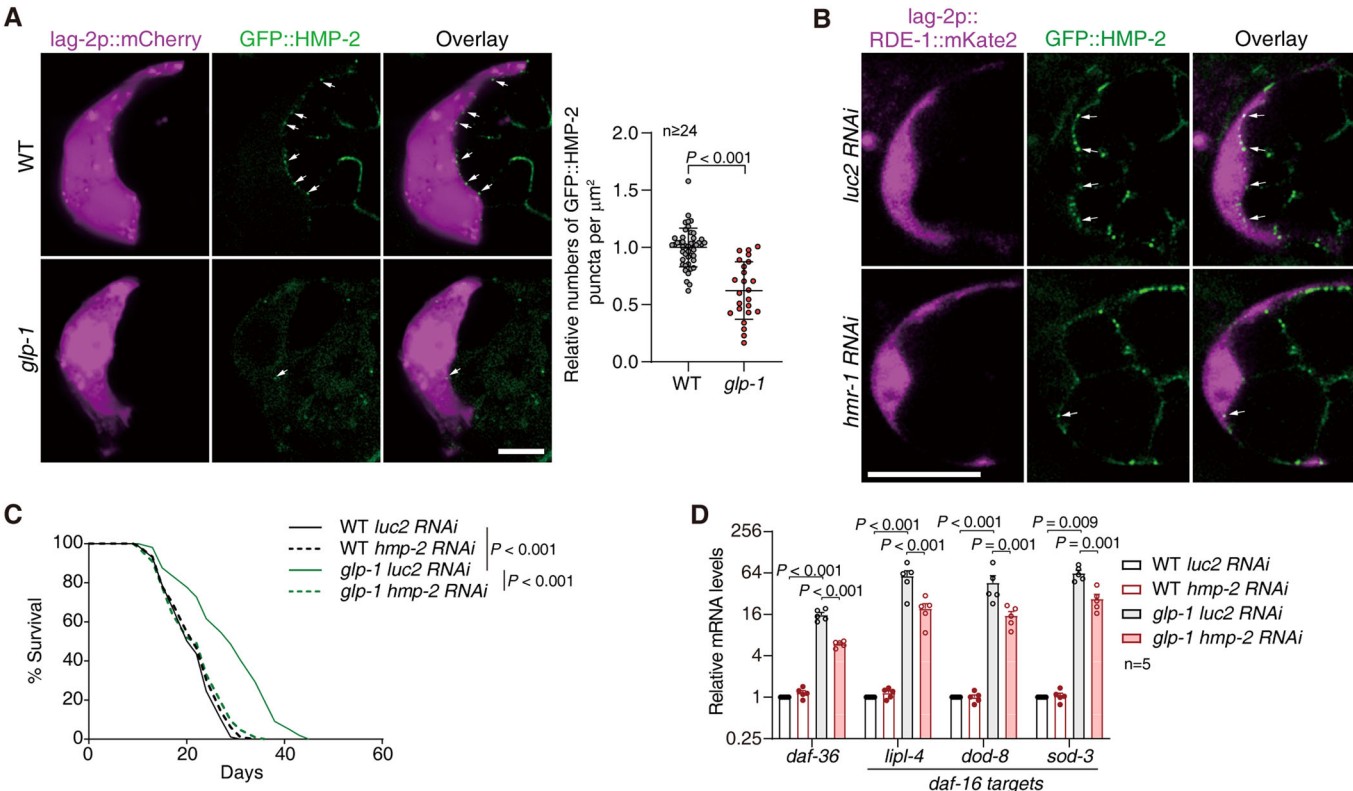

**Figure 4. HMP-2/β-catenin in DTC is required for gonadal longevity.**

(A) GFP::HMP-2 complex on the DTC membrane facing GSC (arrows) is reduced in *glp-1* mutants. The worm ortholog of β-catenin, *hmp-2*, was endogenously tagged with GFP, whereas mCherry was driven by the promoter of a DTC-marker gene, *lag-2*. Representative optical slices focusing on DTC soma are shown to better present the changes in cell adhesion complexes. Scale bar: 5 μm. At least 24 biological replicates (DTCs) were examined. Error bars: SD. (B) The fluorescence of GFP::HMP-2 in DTC upon the indicated DTC-specific RNAi treatments. Arrows denote representative cell adhesions between DTC and GSC. Note that DTC-specific RNAi against *hmr-1* reduced GFP::HMP-2 only in DTC (lag-2p::RDE-1::mKate2) but not in adjacent germline cells. Representative optical slices focusing on DTC soma are shown. Scale bar: 10 μm. (C, D) DTC-specific RNAi against *hmp-2* inhibits longevity (43.8% extension in median lifespan) (C) and the increased levels of indicated genes (D) in *glp-1* mutants. 5 biological replicates were examined. Error bars: SEM. The firefly luciferase gene, *luc2*, serves as the negative control in RNAi assays. Unpaired *t*-test in (A), Mantel-Cox test in (C), paired *t*-test in (D). A representative biological replicate is shown for lifespan analyses in (C). See source data for other biological replicates of (C) and detailed statistics. Source data are available online for this figure.

signalling from L3 effectively alters gonadal longevity (Figs. 2–4). While evolution is unlikely to favour longevity without reproduction, the gonadal longevity signalling from GSC quality control could be important to the survival of the worm population in the wild. When the GSC pool is compromised by harsh environmental conditions, such as food scarcity, the machinery identified in this study could trigger signalling to rescue reproduction and enhance the resistance of the somatic tissues. The stem cell niche maintains stem cells via many other secreted signalling molecules (Chacon-Martinez et al, 2018; Xie, 2008). Whether these factors could also promote longevity will be an exciting issue to explore.

The stem cell niche is highlighted for maintaining the enwrapped stem cells (Chacon-Martinez et al, 2018). Our findings show that DTC controls the longevity of the whole worm, underscoring that niche signalling also regulates other neighbouring tissues. The only stem cells in adult worms are GSCs, whereas mammals have numerous adult stem cells. The degradation of adult stem cell niches is critical in ageing by impairing stem cell-dependent regeneration (Brunet et al, 2022). Our findings suggest that the adult stem cell niche could also modulate ageing by directly

signalling to nearby cells via paracrine or even to distant tissues via endocrine. Since the molecular components discovered in this study are well conserved, it will be interesting to see if a similar mechanism impacts ageing in vertebrates.

## Methods

### *C. elegans* strains and culture

*Caenorhabditis elegans* were grown on NGM plates with standard techniques at 20 °C unless otherwise noted (Brenner, 1974). The strains used in this study are listed in Dataset EV3. Some strains were provided by the CGC, which is funded by NIH Office of Research Infrastructure Programs (P40 OD010440). For synchronisation, eggs laid in the desired time window (4 h to O/N) were collected unless otherwise noted. To remove germline, *glp-1* mutants and corresponding control worms were grown at 25 °C from egg to day 1 of adulthood (D1) and then switched to 20 °C. All worms were examined at D1 unless otherwise noted.

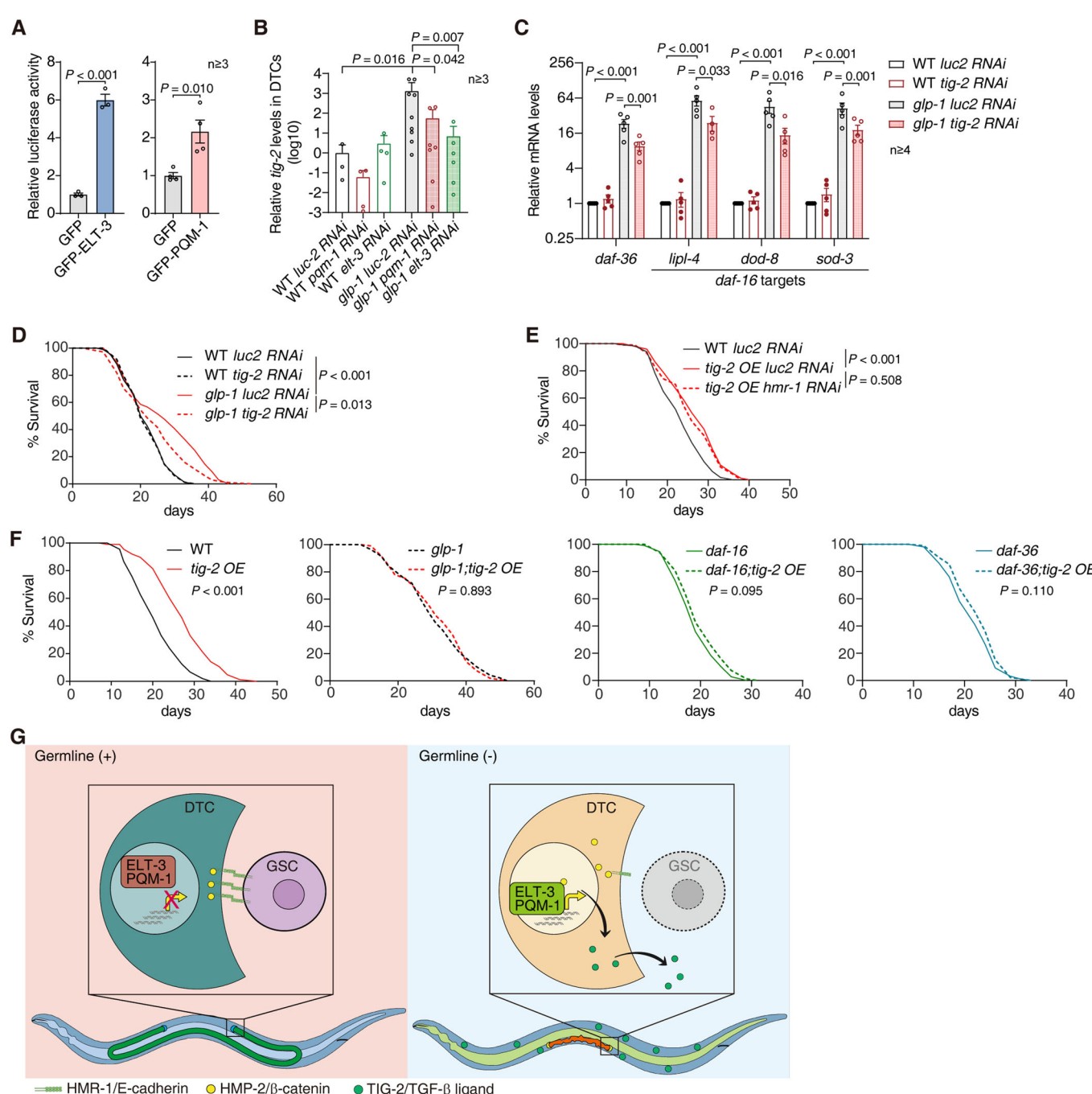

**Figure 5. A TGF-β ligand from DTC is required for gonadal longevity.**

(A) The luciferase reporter driven by *tig-2* promoter is activated by GFP-tagged ELT-3 or PQM-1. Error bars: SEM. At least 3 biological replicates were examined. (B) *tig-2* is upregulated in the DTC of *glp-1* mutants by *elt-3* and *pqm-1*. At least 3 biological replicates were examined. Error bars: SD. (C) Inhibiting *tig-2* expression in DTC suppresses the upregulation of the indicated genes in *glp-1* mutant worms. Error bars: SEM. At least 4 biological replicates were examined. (D) Survival curves of WT worms and *glp-1* mutants under indicated DTC-specific RNAi treatments. DTC-specific inhibition of *tig-2* suppresses the 18.1% extension in median lifespan. (E) DTC-specific RNAi against *hmr-1* does not further extend the lifespan of the worms with *tig-2* overexpressed in DTC. Overexpressing *tig-2* in DTC and suppressing *hmr-1* in DTC extends median lifespan by 14.8% and 10.9%, respectively. (F) Overexpressing *tig-2* in DTC extends the lifespan of WT worms (20.6% extension in median lifespan), but not that of *glp-1*, *daf-16*, or *daf-36* mutants. (G) A graphic summary. The removal of germline disrupts the HMR-1/E-cadherin-based DTC-GSC adhesions, releasing HMP-2/β-catenin from the membrane. HMP-2 controls the transcriptomic changes in DTC through the two GATA TFs, ELT-3 and PQM-1. TIG-2/TGF-β ligand is thus upregulated in DTC and secreted to activate downstream gonadal longevity signalling. See Discussion for details. The firefly luciferase gene, *luc2*, serves as the negative control in RNAi assays. Unpaired *t*-test in (A–C), Mantel–Cox test in (D–F). A representative biological replicate is shown for lifespan analyses in (D–F). See source data for other biological replicates of (D–F) and detailed statistics. Source data are available online for this figure.

## Cell culture and transfection

HEK293T cells (ATCC) were maintained in DMEM medium (Thermo Fisher Scientific, C11995500BT) supplemented with 10% fetal bovine serum (Thermo Fisher Scientific, 10099141) at 37 °C, 5% $CO_2$. Cells were authenticated by morphology and tested for mycoplasma contamination before experiments. HEK293T cells were transfected using Lipomax Transfection Reagent (Sudgen, 32012) of PEI Transfection Reagent (Proteintech, Cat# PR40001) following the manufacturer's instructions.

## Plasmid construction

All plasmids used in this study were constructed by Gibson Assembly. Primers used in plasmid constructions are listed in Dataset EV3.

To generate *L3781-lag-2p::tig-2::mCherry*, 2800 bp of the promoter from pJK590 and synthesised coding sequence of *tig-2* (Xinzhuo Biotech) were PCR and cloned into *L3781-mCherry* (Zhou et al, 2019).

To generate *L3781-lag-2p::mCherry*, 2800 bp of *lag-2* promoter were PCR amplified from N2 genomic DNA and cloned into *L3781-mCherry* (Zhou et al, 2019).

To generate *L3781-lag-2p::mGreenLantern::hmp-2*, 2800 bp of *lag-2* promoter and *hmp-2* PCR amplified from N2 genomic DNA, mGreenlantern amplified from H2B-mGreenLantern were cloned into L3781.

To generate *L3781-vha-6p::tig-2::mCherry.::tig-2 3'UTR*, *tig-2* cDNA was PCR amplified from N2 cDNA and cloned into *L3781-mCherry* (Zhou et al, 2019).

To generate *L3781-tig-2p::tig-2::mCherry*, 2 kb of *tig-2* promoter, *tig-2* cDNA and *tig-2* 3'UTR were inserted into L3781.

To generate *L3781-elt-3p::elt-3::mCherry::elt-3 3'UTR*, 2 kb of *elt-3* promoter and *elt-3* were amplified from N2 genomic DNA and inserted into *L3781-mCherry* (Zhou et al, 2019).

To generate *L3781-pqm-1p::pqm-1::mCherry::pqm-1 3'UTR*, 2 kb of *pqm-1* promoter and *pqm-1* were amplified from N2 genomic DNA and inserted into *L3781-mCherry* (Zhou et al, 2019).

To generate *pCFJ150-lag-2p::rde-1::P2A::mKate2*, *lag-2* promoter of 2982 bp was amplified from pJK590, *rde-1* from pCF1020, and *P2A::mKate2* from TC690.

To generate *pEGFP-elt-3*, *pEGFP-pqm-1*, and *pEGFP-hmp-2*, *elt-3*, *pqm-1*, and *hmp-2* cDNA were amplified from N2 cDNA and cloned at the C terminal, N terminal, and N terminal of *EGFP* in *pEGFP-c1*, respectively.

To generate *elt-3-FLAG* and *pqm-1::FLAG*, EGFP in pEGFP-c1 was first replaced with 3xFLAG tag. cDNA of *elt-3b* and *pqm-1* amplified from N2 cDNA were subsequently inserted at the N-terminal of 3xFLAG tag, respectively.

## Transgenes

*L3781-lag-2p::tig-2::mCherry*, *L3781-lag-2p::mGreenLantern::hmp-2*, *L3781-vha-6p::tig-2::mCherry*, *L3781-elt-3p::elt-3::mCherry::elt-3 3'UTR*, and *L3781-pqm-1p::pqm-1::mCherry::pqm-1* 3'UTR were respectively injected into N2 with an injection marker of *myo-2p::mCherry*, to generate *sydEx298*, *sydEx390*, *sydEx392*, *sydEx289*, and *sydEx282*.

To generate *sydIs135* and *sydIs148, L3781-lag-2p::mCherry* and *L3781-tig-2p::tig-2 cDNA::mCherry::tig-2 3'UTR* were injected into N2, respetively. Plasmid concentrations for microinjections were 50 ng/µl for the genes of interest and 2.5 ng/µl for the injection marker, respectively. The UV integration of *sydIs135* and *sydIs148* and the knock-in of *sydIs3308* were made by SunnyBiotech.

## DTC isolation

For RNA-Seq, worms expressing DTC-specific GFP were anesthetised in 7.5 mM levamisole at day 1 of adulthood. Gonads were dissected with an insulin needle. Dissected gonads were incubated in 20 mg/ml Pronase to segregate DTCs (Merck Millipore, 537088). On an Olympus IX73 microscope, DTCs were isolated from the undigested gonad by an insulin needle and picked by a broken microinjection needle (Harvard Apparatus) mounted on an Eppendorf TransferMan 4r.

For other purposes, worms were lysed, as reported (Zhou et al, 2019). In brief, worms were first lysed in 200 mM DTT, 0.25% SDS, 20 mM HEPES, and 3% sucrose with rocking at 700 rpm for 7 min at 20 °C and subsequently digested in 20 mg/ml Pronase (Merck Millipore, 537088) and 2 mg/ml Collagenase IV (Worthington, LS004186) with rocking at 1200 rpm for 5 min at 20 °C. Digested worms were transferred into PBS on a glass slide and subjected to DTC isolation by micromanipulation with an Eppendorf TransferMan 4r mounted on an Olympus IX73 microscope.

Isolated DTCs were transferred into 200 µl PBS with 0.1% BSA and further collected into ~4 µl PBS with 0.1% BSA. About 12 DTCs were collected for each sample. The sample purity was validated by RT-qPCR of tissue-specific genes.

## RT-qPCR

For RT-qPCR, more than 100 well-fed synchronised worms were collected into TRIzol Reagent (Invitrogen, 15596018) and column purified by RNeasy Mini (QIAGEN, 74104). cDNA was subsequently generated by TaKaRa PrimeScript™ RT reagent Kit (Takara, RR037A). Quantitative RT-PCR was performed with NovoStart® SYBR qPCR SuperMix Plus (Novoprotein, E096) on a QuantStudio™ 6 Flex Real-time PCR System (Applied Biosystems) or a CFX384 Touch™ Real-Time PCR Detection System (Bio-Rad). A combination of *ama-1* and *cdc-42* was used as reference. At least three biological replicates, with four technical replicates in each, were examined. Primer sequences are listed in Dataset EV3 or as reported (Johnson et al, 2014; Kaplan et al, 2015; Zhou et al, 2019).

For RT-qPCR of isolated DTCs, cDNA was generated using TaKaRa PrimeScript™ RT reagent Kit (Takara, RR037A) as reported (Picelli et al, 2013). cDNA was further amplified for 29 cycles with Q5 (NEB, M0491) and diluted 3 times for qPCR.

## RNA-Seq of DTCs

Isolated DTCs were collected in RNase-free PCR tubes containing single-cell RNA lysis buffer with 0.45% (v/v) NP40. For each sample, 6 isolated DTCs were pooled together. Reverse

transcription was performed using SuperScript II reverse transcriptase (Invitrogen, 18064014). cDNA was further PCR amplified with KAPA HiFi HotStart ReadyMix (KAPA Biosystems, KK2601) for 18 cycles. Tissue contamination and sample quality were examined by RT-qPCR of tissue-specific genes and housekeeping genes, respectively. Uncontaminated samples were subjected to automated single-cell RNA-Seq library construction based on the Bravo robot station (Peng et al, 2019). In brief, the PCR product was purified using 0.8x AMPure XP beads (Agencourt, A63881) and quantified with Qubit dsDNA HS Assay Kit (Thermo Fisher Scientific, Q32856) on an Envision® Multilabel Plate Reader (PerkinElmer). cDNA library was constructed by Nextera XT DNA Library Preparation Kit (Illumina, FC-131-1096) and sequenced on an Illumina NovaSeq 6000 Sequencing System using a 150 bp paired-end-reads setting.

## RNA-Seq data analysis

For data analysis, reference genome sequences and gene annotation were downloaded from ENSEMBL (WBcel235). Raw data were pre-processed by Fastp (v0.12.1) (Chen et al, 2018) with default parameters. Cleaned data were then aligned to the reference genome via the software Hisat2 (v2.1.0) (Kim et al, 2015). FeatureCounts (v1.5.3) was used to count the reads mapped to each gene (Liao et al, 2014). The Differentially Expressed Genes (DEGs) were calculated by Deseq2 (v1.26.0) with padj <0.05 and fold of change >2 (Love et al, 2014).

To discover transcription factor binding motifs in the DEG promoters, start codons' positions of these genes were extracted and analysed by findMotifsGenome.pl of HOMER (v4.11) based on *C. elegans* genome sequence of WBcel235 and with a parameter of '-size -600, +50 -len 8' (Heinz et al, 2010). AnnotatePeaks.pl of HOMER was used to find individual motif occurrences in these promoters.

Gene set enrichment analysis was performed using WormCat (http://wormcat.com/) at its default settings (Holdorf et al, 2020). Significance scores were as Fisher's exact test *P* values. Terms were considered significant if the *P* value < 0.05.

## RNA interference

RNAi experiments were performed as described (Kamath et al, 2001). For DTC-specific RNAi, synchronised worms were grown on OP50 plates until L3 and transferred to corresponding RNAi plates. For DTC-specific double RNAi, equal amounts of RNAi bacteria were mixed. The RNAi constructs against *hmr-1*, *hmp-2*, *tig-2*, GFP or mNeonGreen (mNG) were prepared in this study. The RNAi construct against a firefly luciferase (*luc2*) was used as control. The strain of HT115 [*L4440::luc2*] is a gift from Antebi lab in MPI-AGE. The strains of HT115 [*L4440::pqm-1*] and HT115 [*L4440::elt-3*] are gifts from Cai lab in ION, CAS.

## Laser ablation

To ablate the worm germline, worms at L1 were mounted on 5% agar pads and anesthetised using 7.5 mM levamisole. Laser ablation of Z2 and Z3 cells were performed on an Olympus SpinSR microscope. Worms post laser ablation were washed back onto plates with M9.

## Lifespan assays

Adult lifespan analysis was performed as reported (Gerisch et al, 2007). Worms were transferred to fresh plates every other day until day 10 of adulthood to avoid contamination from progenies and subsequently monitored every other day for dead worms. Worms undergoing internal hatching, bursting vulva, or crawling off the plates were censored. Statistical analysis was performed with the Mantel-Cox log-rank test.

## Microscopy

Worms were mounted on 5% agar pads and anesthetised using 7.5 mM levamisole. The images were captured using a Leica TCS SP8 to validate the effect of DTC-specific RNAi, a Zeiss LSM880 Ariyscan for HMR-1::GFP and GFP::HMP-2, an Olympus SpinSR for mGL::HMP-2, an Olympus FV3000 for the expression of *lag-2p::tig-2::mCherry*, and an Olympus BX53 microscope for other assays.

To quantify HMR-1::GFP and GFP::HMP-2 puncta, a Z-stack confocal images focusing on DTC soma were used to re-constitute DTC soma and part of its protrusion in 3D by Imaris (Oxford Instruments). The DTC membrane facing GSC was unbiasedly identified from the reconstituted DTC soma by Imaris. The puncta on this membrane were identified and counted by Imaris.

Fluorescence intensities were measured by Image J (NIH), with the background signal subtracted as reported (Shen et al, 2012). For the nuclear HMP-2 enrichment in DTC, the DTC nucleus was selected and the fluorescence intensity of mGL::HMP-2 was measured in the selected area.

## Luciferase reporter assay

HEK293T Cells were collected and examined for luciferase activity 48 h post-transfection by Dual-Luciferase® Reporter Assay System (Promega) on an EnVision® Multilabel Plate Reader (PerkinElmer), as the manufacturers instructed. At least three independent assays were performed.

## Co-immunoprecipitaton

48 h post transfection, cells were washed by PBS and collected into lysis buffer (Beyotime, P0013) with PMSF (Roche, 10837091001) and protease inhibitor (Merck, 539134) freshly added. After lysis, 30 μl of the supernatant was collected as input. The rest was incubated with Anti-GFP Affinity Beads (Smart-Lifesciences, SA070001) or Anti-DYKDDDDK Affinity beads (Smart-Lifesciences, SA042001) for 2–4 h at 4 ℃. Afterwards, the beads were washed with lysis buffer and wash buffer (20 mM Tris, 150 mM KCl, 0.5% NP-40, 1 mM EDTA, 10% glycerol, 10 mM Na Pyrophosphate, pH 7.5) with protease inhibitor freshly added. Immunoprecipitated proteins were eluted by 2x SDS loading buffer or Gly-HCl (pH 3.0).

## Western blot

Proteins were separated by reducing SDS-PAGE and transferred to nitrocellulose membranes. Membranes were blocked by 5% defatted milk and then blotted with primary antibodies against

GFP (1:2000, Santa Cruz, sc-9996, or Proteintech, 50430-2-AP), FLAG (1:2000, Sigma, F3165, F7425). HRP-conjugated secondary antibodies against mouse or rabbit IgG (1:5000, Life Technologies, G-21040, G-21234) were subsequently used. Signals of western blotting were captured by a Tanon™ 5200 Chemiluminescent Imaging System, and analysed using Adobe Photoshop with background signals subtracted as reported (Zhang et al, 2021). Uncropped blots are shown in source data.

### Chromatin immunoprecipitation

HEK293T cells transfected with pEGFP, pEGFP-ELT-3, or pEGFP-PQM-1, with tig-2p::luciferase were subjected to chromatin immunoprecipitation (ChIP) 48 h post-transfection, as reported (He et al, 2021; Lee et al, 2006). In brief, cells were crosslinked by incubating with 1% paraformaldehyde for 10 min at room temperature. An incubation with 125 mM glycine was used to quench the reaction. Afterwards, cells were collected in HLB (50 mM HEPES-KOH, 150 mM NaCl, 1 mM EDTA, 0.1% sodium deoxycholate, 1% Triton X-100, 0.1% SDS, 1 mM PMSF, protease inhibitor cocktail, pH 7.5) and incubated on ice for 30 min. Cells were subsequently sonicated by a Q800R3 (Qsonica). Sonicated cell lysates were aliquoted as input. Lysates with 10 μg DNA was incubated overnight with 3 μl ChIP grade GFP antibody (Abcam, Cat# ab290) at 4 °C and immunoprecipitated using 30 μl ChIP-Grade Protein G Agarose Beads (CST, 9007S) for 2 h at 4 °C. Beads were washed with WB1 (50 mM HEPES-KOH, 150 mM NaCl, 1 mM EDTA, 1% sodium deoxycholate, 1% Triton X-100, 0.1% SDS, 1 mM PMSF, pH 8.0), WB2 (50 mM HEPES-KOH, 1 M NaCl, 1 mM EDTA, 0.1% Na deoxycholate, 1% Triton X-100, 0.1% SDS and 1 mM PMSF, pH 8.0), WB3 (50 mM Tris, 0.25 mM LiCl, 1 mM EDTA, 0.5% NP-40 and 0.5% Na deoxycholate, pH 8.0), and TE. After washing, TF-chromatin complex was eluted with 150 μl elution buffer (1% SDS, 100 mM NaHCO$_3$). Immunoprecipitated complex was digested by RNase A (Sigma, EN0531) and proteinase K (NEB, P8107S), and then purified using StarPrep PCR & DNA Fragment Purification Kit (GenStar, D206-04). Purified DNA was analysed by qPCR. The cDNA (211-277) of luciferase served as the negative control (NC). The primers used are listed in Dataset EV3. The signal of the promoter region in immunoprecipitants relative to that in input was compared.

### Statistical analysis

Tukey correction was used in the comparison of multiple samples. Statistical tests in RT-qPCR assays were from the dCt values. Statistical tests were performed as indicated using GraphPad Prism (GraphPad software), unless otherwise noted. The changes with *P* values smaller than 0.05 are considered as statistically significant. *P* values calculated by GraphPad were rounded to three decimal places. Sample size was determined by referring to well-accepted previous studies. The investigators were blinded to the genotypes and dsRNA treatment. Detailed information on statistics is shown in source data.

## Data availability

The DTC-specific RNA-Seq data from this publication have been deposited to the GEO database (https://www.ncbi.nlm.nih.gov/geo)

with the accession ID of GSE213802 (https://www.ncbi.nlm.nih.gov/geo/query/acc.cgi?acc=GSE213802). All other data and non-commercial materials and reagents are available from the corresponding author upon reasonable request.

The source data of this paper are collected in the following database record: biostudies:S-SCDT-10_1038-S44318-024-00185-3.

## Peer review information

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

## Acknowledgements

The authors thank the institutional core facilities for cell biology and molecular biology for instrumental and technical support, Drs Adam Antebi (MPI-AGE), Shiqing Cai (ION, CAS), E Jane Albert Hubbard (NYU) for strains, Dr Minshu Zhang (PKU) for the cDNA of mGreenLantern, Dr Yun Liu and Mr Hua Sun (Zeiss) for support of microscopy. This research was supported by National Natural Science Foundation of China (32070731).

## Author contributions

**Meng Liu**: Conceptualization; Data curation; Formal analysis; Investigation; Visualization; Writing—original draft; Writing—review and editing. **Jiehui Chen**: Data curation; Formal analysis; Investigation. **Guizhong Cui**: Data curation; Formal analysis; Investigation. **Yumin Dai**: Data curation; Formal analysis; Investigation. **Mengjiao Song**: Data curation; Formal analysis; Investigation. **Chunyu Zhou**: Data curation; Formal analysis; Investigation. **Qingyuan Hu**: Data curation; Formal analysis; Investigation. **Qingxia Chen**: Data curation; Formal analysis; Investigation. **Hongwei Wang**: Data curation; Formal analysis; Investigation. **Wanli Chen**: Data curation; Formal analysis; Investigation. **Jingdong Jackie Han**: Data curation; Supervision. **Guangdun Peng**: Data curation; Formal analysis; Supervision; Investigation. **Naihe Jing**: Data curation; Supervision. **Yidong Shen**: Conceptualization; Data curation; Formal analysis; Supervision; Funding acquisition; Investigation; Methodology; Writing—original draft; Project administration; Writing—review and editing.

Source data underlying figure panels in this paper may have individual authorship assigned. Where available, figure panel/source data authorship is listed in the following database record: biostudies:S-SCDT-10_1038-S44318-024-00185-3.

## Disclosure and competing interests statement

The authors declare no competing interests.

# Expanded View Figures

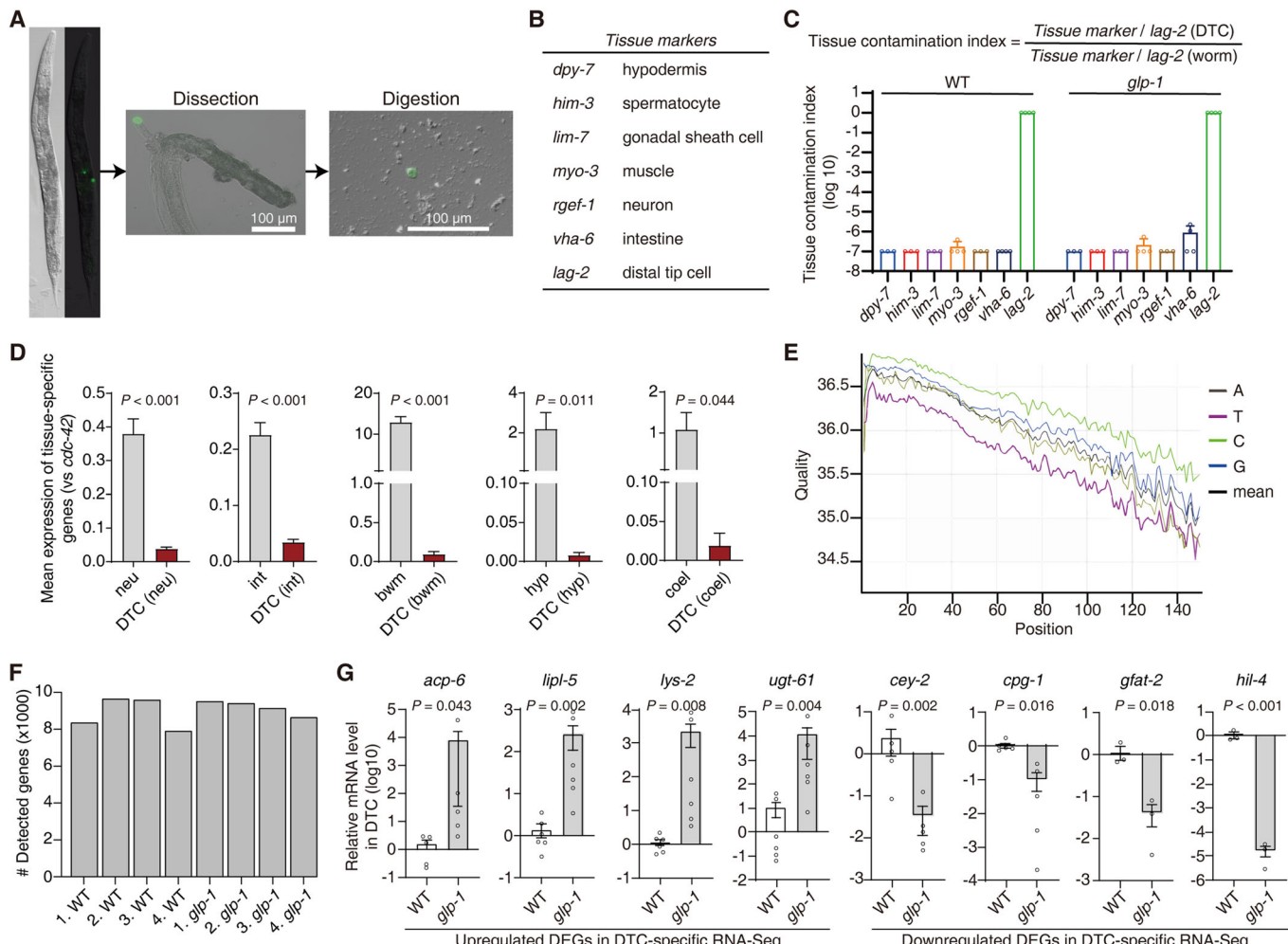

**Figure EV1. RNA-Seq of the isolated DTCs from WT worms and *glp-1* mutants.**

(A) The workflow of isolating GFP-labelled DTCs from worms. Scale bar: 100 μm. (B) A list of tissue marker genes. (C) RT-qPCR analysis for the purity of isolated DTCs subjected to RNA-Seq. When gene expression was below the detection limit, the corresponding tissue contamination index was set as $10^{-7}$. At least 4 biological replicates were examined. Error bars: SD. (D) The expression of genes detected specifically in neuron (neu), intestine (int), body wall muscle (bwm), hypodermis (hyp), and coelomocyte (coel) by RNA-Seq (Wang et al, 2022) and in our DTC-specific RNA-Seq dataset. *cdc-42* serves as a reference gene for normalization. 345, 140, 33, 24, and 4 genes specifically detected in neu, int, bwm, hyp, and coel are analysed by their mean expression levels. Error bars: SEM. Unpaired *t*-test. At least 4 biological replicates were examined. (E) Reads quality of RNA-Seq on isolated DTCs. Quality shows the error rate of indicated bases. The higher the base quality value is, the less likely the base is mis-detected. The error rate corresponding to quality 30 is 99.9%, and 40 is 99.99%. (F) Detected genes in the indicated DTC samples. (G) RT-qPCR of isolated DTCs for the representative upregulated and downregulated genes identified by DTC-specific RNA-Seq. Unpaired *t*-test. At least 3 biological replicates were examined. Source data are available online for this figure.

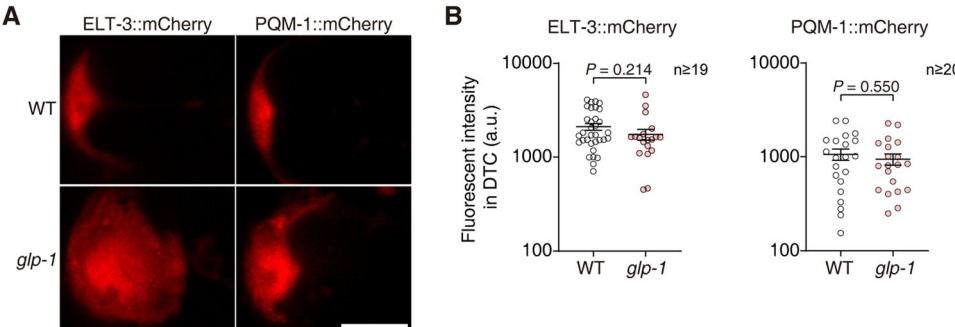

**Figure EV2. The loss of germline does not change the expression of ELT-3 or PQM-1 in DTC.**

(A) The expression of ELT-3::mCherry or PQM-1::mCherry in the DTC of indicated strains. Representative optical slices focusing on DTC soma are shown. Scale bar: 5 μm.
(B) Quantification of the fluorescent intensity of ELT-3::mCherry and PQM-1::mCherry in the DTC of indicated strains. Error bars: SEM. Unpaired *t*-test. At least 19 biological replicates (DTCs) were examined. Source data are available online for this figure.

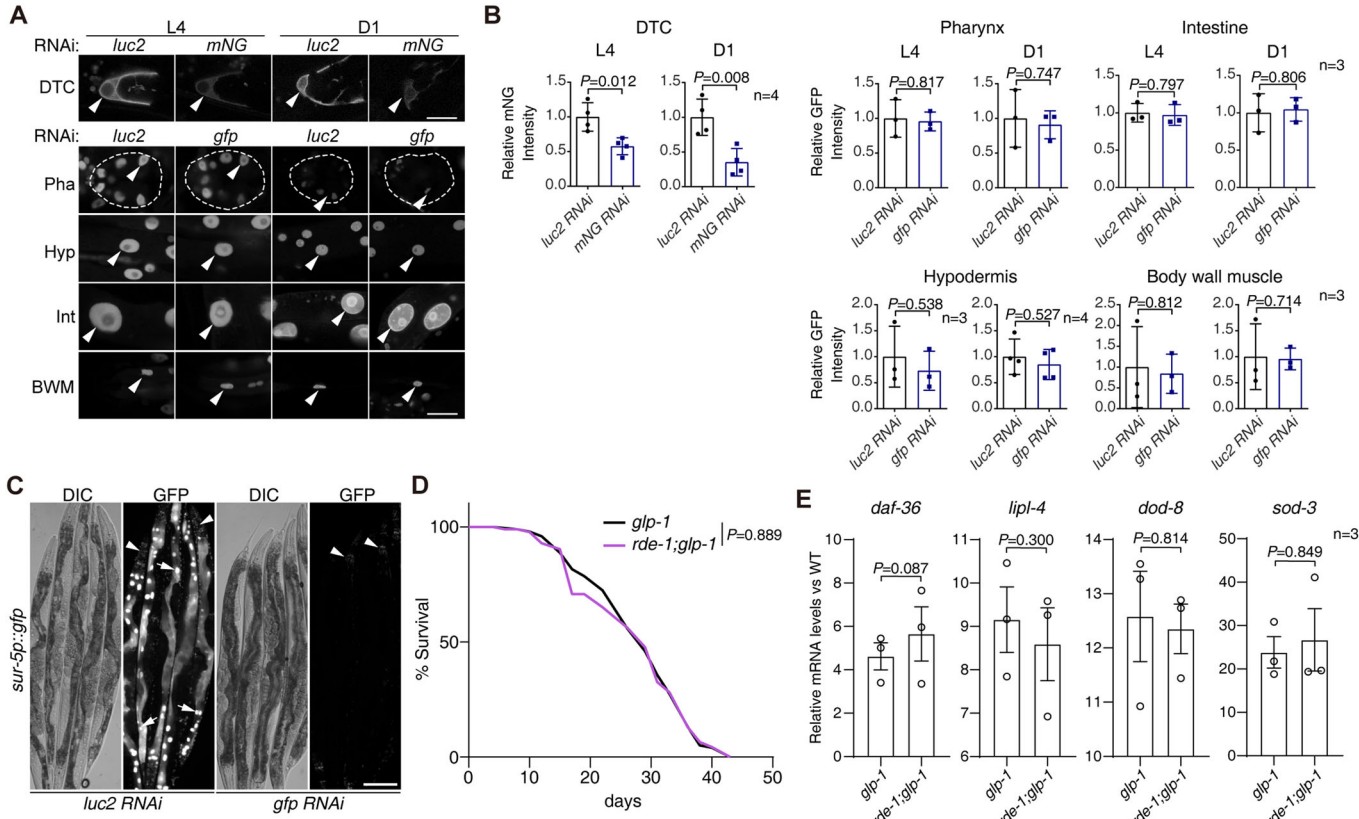

**Figure EV3. The validation of the specificity of RNAi in DTC.**

(A) Representative images of RNAi against mNG in DTC and GFP in other indicated tissues in the strain of DTC-specific RNAi. The strain for DTC-specific RNAi is a *rde-1* mutant with *rde-1* rescued in DTC via a single copy transgene, as reported by Sherwood lab. mNG was driven by the promoter of *lag-2*, whereas GFP was by the promoter of *sur-5*. Arrowheads denote the tissues of interest. L4: the 4th larval stage, D1: day 1 of adulthood. Scale bar: 10 µm. (B) The fluorescent intensity of mNG in DTC and GFP in other tissues upon indicated DTC-specific RNAi treatment. Error bars: SD. Unpaired *t*-test. At least 3 biological replicates were examined. (C) RNAi against *gfp* efficiently reduces the expression of *sur-5p::GFP* in WT worms. Note that the GFP signal in neurons are barely affected by *gfp* RNAi due to the insensitivity of RNAi in this tissue. Arrows denote intestine, arrowheads denote head neurons. Scale bar: 100 µm. (D) The survival curves of *glp-1* and *rde-1;glp-1* mutants. Mantel-Cox test. A representative biological replicate is shown for lifespan analyses. See source data for other biological replicates and detailed statistics. (E) The transcription of indicated genes in *glp-1* and and *rde-1;glp-1* mutants. Error bars: SEM. Unpaired *t*-test. 3 biological replicates were examined. The firefly luciferase gene, *luc2*, serves as the negative control in RNAi assays. Source data are available online for this figure.

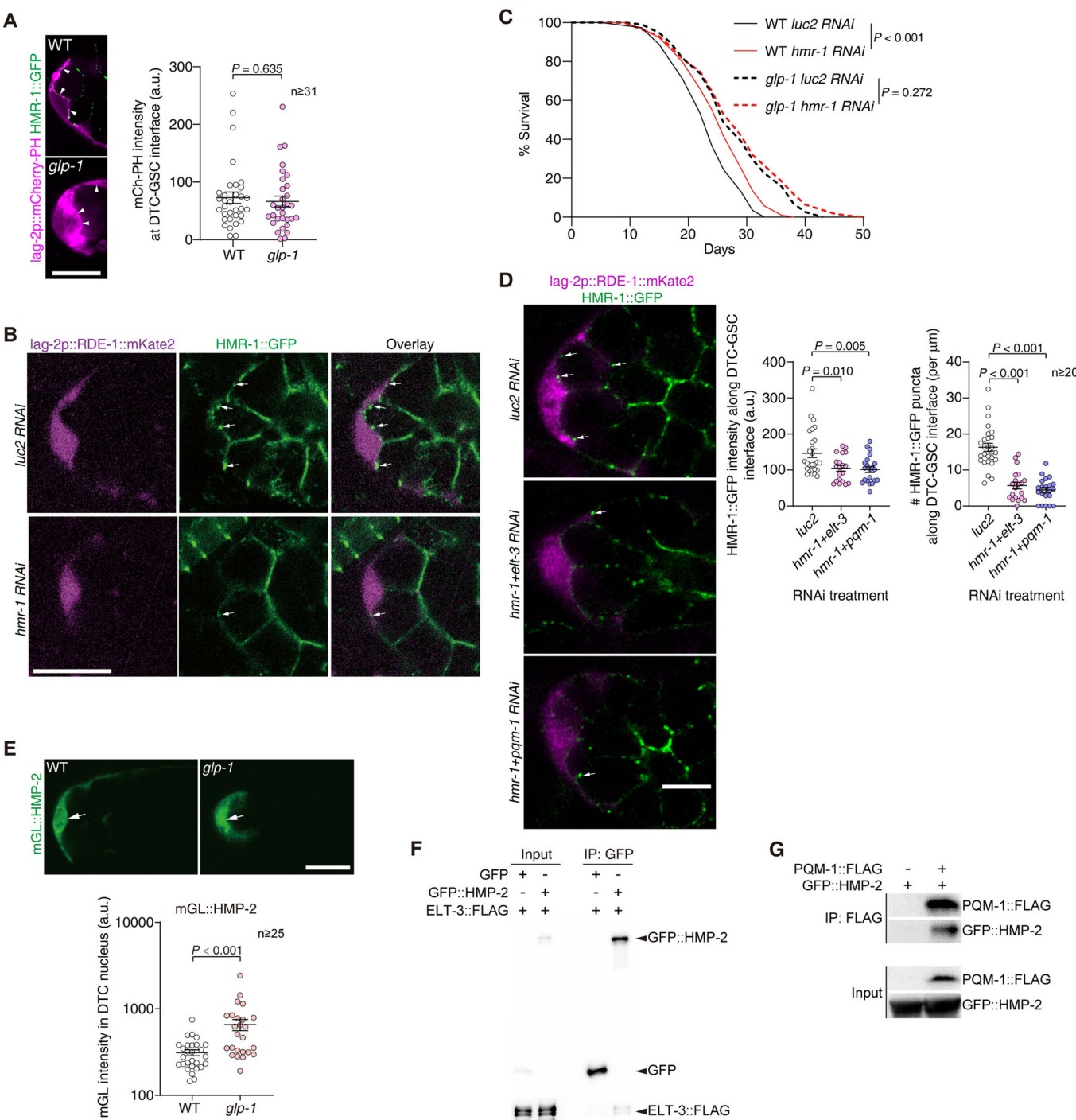

**Figure EV4. The reduction of cell adhesions between DTC and GSC induces gonadal longevity.**

(A) The membrane bound mCherry-PH is not decreased at DTC-GSC interface (arrowheads), unlike HMR-1::GFP labelled cell adhesions. Two optical slices from the Z-stack in Fig. 3B are shown. Unpaired *t*-test. Error bars: SEM. Scale bar: 10 μm. At least 31 biological replicates (DTCs) were examined. (B) DTC-specific RNAi against *hmr-1* reduced DTC-GSC adhesions (arrows) but not the adhesions in other germline cells. lag-2p::RDE-1::mKate2 labels DTC. Representative optical slices focusing on DTC soma are shown. Scale bar: 10 μm. (C) DTC-specific RNAi against *hmr-1* extends the lifespan of WT worms (8.7% extension in median lifespan) but not that of *glp-1* mutants. Mantel-Cox test. A representative biological replicate is shown. See source data for other biological replicates and detailed statistics. (D) HMR-1::GFP signal along the DTC-GSC interface and DTC-GSC adhesions (arrows) are significantly reduced upon the DTC-specific double RNAi against *hmr-1* and indicated GATA TFs (same as Fig. 3E, F). Representative optical slices focusing on DTC soma are shown. Scale bar: 5 μm. Unpaired *t*-test. Error bars: SEM. At least 20 biological replicates (DTCs) were examined. (E) The nuclear localization of mGreenLantern (mGL)-tagged HMP-2 (arrows) is increased in the DTC of *glp-1* mutants. Representative optical slices focusing on DTC soma are shown. Scale bar: 10 μm. Error bars: SEM. Unpaired *t*-test. At least 25 biological replicates (DTCs) were examined. (F, G). Co-immunoprecipitation of ELT-3::FLAG (F) or PQM-1::FLAG (G) with GFP::HMP-2 in HEK293T cells. The firefly luciferase gene, *luc2*, serves as the negative control in RNAi assays. Source data are available online for this figure.

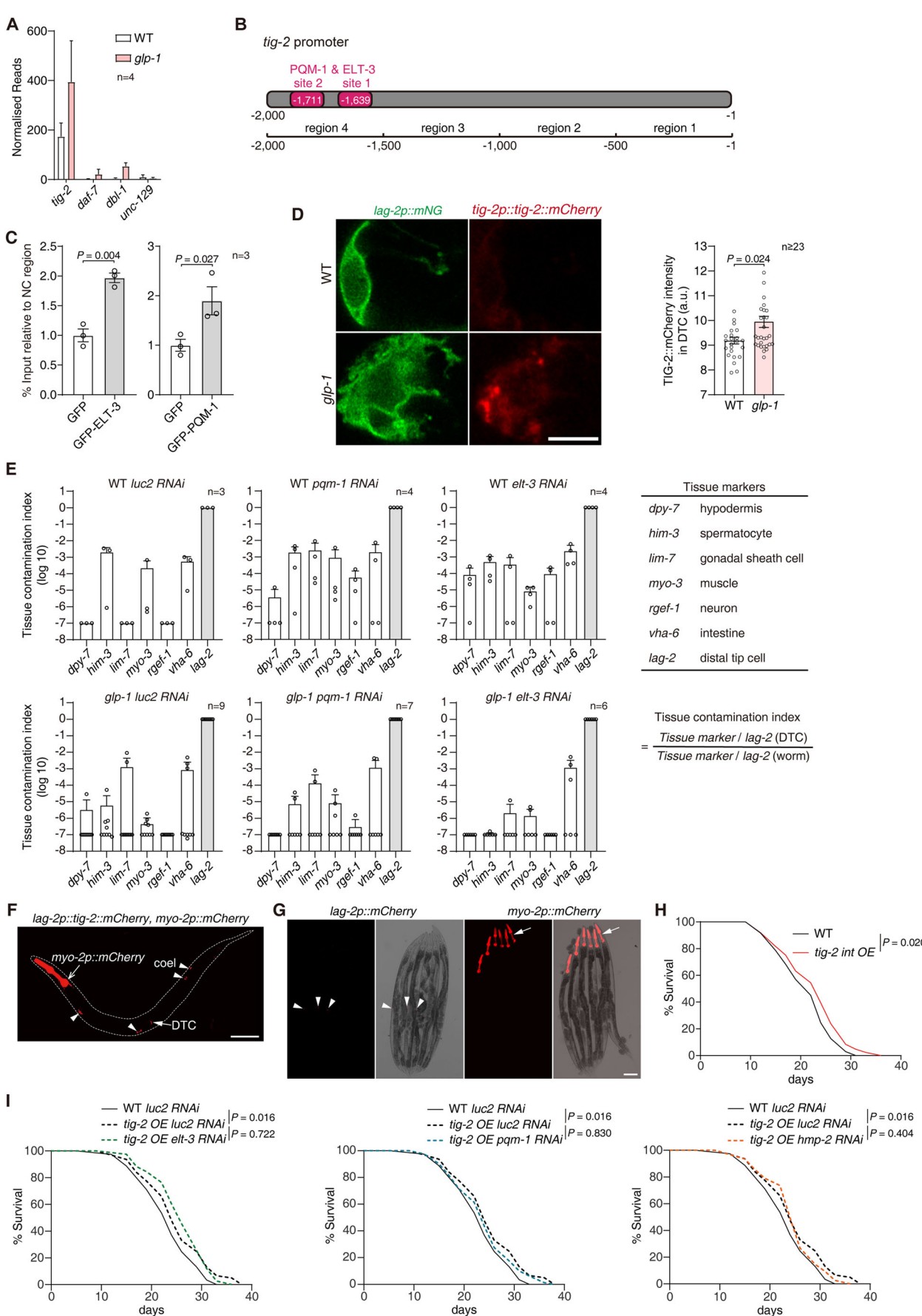

◀  **Figure EV5.   The induction of *tig-2* in DTC upon germline removal promotes longevity.**

(A) The expression level of the four detected TGF-β ligand genes in DTC-specific RNA-Seq. *n*: the number of biological replicates. Each sample of one biological replicate contains around 12 DTCs. Error bars: SEM. (B) A diagram showing the binding sites of ELT-3 and PQM-1 in *tig-2* promoter. (C) HEK293T cells co-expressing indicated GFP proteins and a luciferase reporter driven by *tig-2* promoter were subjected to ChIP analysis. Note that *tig-2* promoter is co-immunoprecipitated with GFP-ELT-3 and GFP-PQM-1. The 3′-UTR of luciferase serves as the negative control. 3 biological replicates were examined. Error bars: SEM. (D) The expression of TIG::mCherry in DTC is increased in *glp-1* mutants. lag-2p::mNG labels DTC. Representative optical slices focusing on DTC soma are shown. Scale bar: 5 μm. At least 23 biological replicates (DTCs) were examined. Error bars: SEM. (E) Isolated DTCs for RT-qPCR of *tig-2* were analysed for their purity. When gene expression was below the detection limit, the corresponding tissue contamination index was set as $10^{-7}$. At least 3 biological replicates were examined. Error bars: SD. (F, G) The expression of TIG-2::mCherry (F) and mCherry (G) by indicated promoters. The strong red fluorescence in the pharynx (F) is from the injection marker, *myo-2p::mCherry*. Note in (F) that TIG-2::mCherry is detected not only in DTC (arrow) but also in coelomocytes (arrowheads). mCherry driven by *lag-2* or *myo-2* promoter was not detected in coelomocyte. Arrowheads in (G) denote DTC, whereas arrows in (G) pharynx. Scale bar: 100 μm. (H) The artificial overexpression of *tig-2* in intestine extends the lifespan of WT worms (9.1% extension in median lifespan). (I) DTC-specific RNAi against *elt-3, pqm-1*, or *hmp-2* does not suppress the longevity induced by overexpressing *tig-2* in DTC (10.3% extension in median lifespan). The firefly luciferase gene, *luc2*, serves as the negative control in RNAi assays. Unpaired *t*-test in (C) and (D), Mantel-Cox test in (H) and (I). A representative biological replicate is shown for lifespan analyses in (H) and (I). See source data for other biological replicates and detailed statistics of (H) and (I). Source data are available online for this figure.

                                                      