## [Peer Review File · The EMBO Journal]

Germline loss in *C. elegans* enhances longevity by disrupting adhesion between niche and stem cells

Yidong Shen, Meng Liu, Jiehui Chen, Guizhong Cui, Yumin Dai, Mengjiao Song, Chunyu Zhou, Qingyuan Hu, Qingxia Chen, Hongwei Wang, Wanli Chen, Jing-Dong Han, Guangdun Peng, and Naihe Jing

Corresponding author: Yidong Shen (yidong.shen@sibcb.ac.cn)

Review Timeline:

Submission Date:	26th Jan 24
Editorial Decision:	23rd Feb 24
Revision Received:	23rd May 24
Editorial Decision:	20th Jun 24
Revision Received:	3rd Jul 24
Accepted:	12th Jul 24

Editor: Daniel Klimmeck

Transaction Report:

This manuscript transferred to The EMBO Journal following peer review at another journal. The peer review comments and authors' responses were made available as agreed with the authors and the other journal, and were taken into account for the decision process at The EMBO Journal.

We appreciate the reviewer's time and efforts on our manuscript. Please see our point-to-point responses highlighted in blue.

Reviewers' comments:

Reviewer #1 (Remarks to the Author):

While the authors have made efforts to address some of the concerns of this reviewer, many of the most critical points have not been addressed.

We are glad to learn that our efforts have been acknowledged by Reviewer #1. We are confident that the remaining concerns about those most critical points are already addressed.

Our detailed responses to Reviewer #1 remaining concerns are listed below:

The quality of the images have not been improved,

-- Reviewer #1's prior comment criticised our DTC images as 'They fail to display the very elaborate, dendritic structure of the DTCs' and cited the recent eLife publication (<https://doi.org/10.7554/eLife.75497>) from the Gordon lab as an example of high-quality DTC images (Li, Singh et al., 2022) (Fig. 1A for reviewers).

We then obtained the same strain used in Gordon lab's paper to label DTC and reimaged DTC (Supplementary Fig. 5a and Movie 2) (Fig. 1B for reviewers). The dendritic structures are clearly seen in WT worms and depleted in *glp-1* mutants, as shown in these Z-stack confocal images. We believe that these new results are of comparable quality to those from Gordon's lab and by the standards of the field. As our study focus on the interactions between DTC and GSC, our images do not include the distal ends of those dendritic structures which enwrap other germ cells (Li et al., 2022).

A Fig. 2g (Li et al., eLife, 2022)
lag-2p::mCherry-PH

B Supplementary Fig. 5a (our study)
lag-2p::mCherry-PH HMR-1::GFP

Fig.1 for reviewers. Z-stack DTC images in the recent eLife publication by Gordon lab (A) and our study (B)

DTC is labelled in cyan by the same transgene in both studies. Arrows in our image denote adhesions on the dendritic structures of DTC. Please also see Supplementary Movie 2. Scale bars: 10 μm (A and B). The DTC in Li et al.'s report is of WT worms. Our images in Supplementary Fig. 5a are now reorganized in Fig. 3B.

As Reviewer #3 suggested, we have moved these data from Supplementary Fig. 5a to Fig. 3B to present our study more straightforwardly.

Meanwhile, we also kept our old DTC images, such as those in Fig. 3G. These images are optical slices focusing on the cell body of DTC from Z-stack confocal images. The maximum projection of the whole Z-stack also shows DTC with many dendritic structures in WT worms (Fig. 2 for reviewers, arrows). It is the shown optical slices but not our imaging that miss the dendritic structures.

Fig. 2 for reviewers. Representative DTC morphologies

in WT worms and *glp-1* mutants. The maximum projection of Z-stack images is shown here. *glp-1* is a genetic mutant of germline ablation. Two WT DTCs and two *glp-1* DTCs are shown. Note that DTC protrusions (arrows) are significantly retracted upon germline ablation.

These optical slices of DTC soma are clearer to show the changes of DTC-GSC adhesions. Upon germline removal, the dendritic structures decrease remarkably, whereas DTC soma is much less affected (Fig. 3B and Supplementary Movie 2). Therefore, we think it will facilitate the readers to recognise and us to quantify the changes of cellular adhesions on DTC when focusing on the less-affected soma.

As our prior rebuttal letter explained, we agree with Reviewer #1 that those dendritic structures provide a critical interface between DTC and GSCs. Given the remarkable deformation of these structures upon germline removal, our previous analysis could underestimate the change in DTC-GSC adhesions. Indeed, there are many HMR-1::GFP labelled adhesions in these dendritic structures in WT worms, as shown in our new data (Fig. 3B and Supplementary Movie 2). The adhesions on dendritic structures are remarkably decreased with the reduction of dendritic structures in *glp-1* mutants.

However, it is hard to define the DTC-GSC interface at these structures precisely because they enwrap not only GSCs at their proximal part, but also other germline cells at their distal ends (Li et al., 2022). Therefore, when quantifying cell adhesions between the two types of cells, we still focused on the DTC-GSC interface at DTC soma for a more precise, although underestimated, analysis (e.g., Fig. 3B). All our images are accordingly focusing on the area around DTC soma.

the authors did not address the question of the biological significance of the reduced density of adhesion

-- Our study is about how reducing DTC-GSC adhesions induces gonadal longevity. Fig. 3C-F shows that disrupting adhesion (*hmr-1* RNAi) significantly affects lifespan and gonadal longevity genes expression. Supplementary Fig. 5f shows the reduction of GFP::HMP-2/ β -catenin on membrane upon DTC-specific RNAi against *hmr-1*.

These are clear evidence showing the biological significance of the reduced density of adhesion within the scope of this study.

Panels c-f from Fig. 3

(c) DTC-specific RNAi against *hmr-1* extends lifespan. Mantel-Cox test.

(d) Inhibiting *hmr-1* in DTC upregulates the expression of indicated genes in WT worms but not in *glp-1* mutants. Error bars: SEM.

(e and f) *elt-3* or *pqm-1* in DTC is required for the extended lifespan (e) and increased expression of *daf-36* and *daf-16* targets (f) induced by DTC-specific RNAi against *hmr-1*.

Panel f from Supplementary Fig. 5

DTC-specific RNAi against *hmr-1* reduced GFP::HMP-2 only in DTC (lag-2p::RDE-1::mKate2) but not in adjacent germline cells. **An optical slice focusing on DTC soma is shown.** Scale bar: 10 μ m.

and did not make efforts to characterise the cell shape and cytoskeletal changes according to the standards of the field.

-- As in Reviewer #1's prior comments, cell shape and cytoskeletal changes are supporting evidence to strengthen our findings that adhesions are reduced in DTC when the germline is removed.

Our new data in Supplementary Fig. 5a and Movie 2 show the change in cell shape, highlighting the reduction of dendritic protrusions of DTC in *glp-1* mutants. We think these images are to the standards of the field, as we explained above. We also described and discussed these observations in the revised main text on page 7-8, and in our prior rebuttal letter on pages 9-10.

Similar phenotypes have been reported by Sherwood lab (Linden, Gordon et al., 2017). We apologise for missing this report and have properly referenced it in our manuscript. Therefore, the change in DTC morphology when the germline is removed is a solid observation and well supports our findings of the reduced adhesions.

Panel a from Supplementary Fig. 5

The morphology of DTC (lag-2p::mCherry-PH) undergo a remarkable change when germline is removed in *glp-1* mutants. Two representative Z-stacking of confocal microscopic images are shown. Scale bar: 10 μ m. **Also see Supplementary Movie 2.**

Since the previous study by Linden et al. has already measured the change in the size of DTC cap and plexus (Linden et al., 2017), we further quantified the changes in the numbers of dendritic structures last week. The dendritic protrusions from DTCs are significantly reduced in *glp-1* mutants (Fig. EV5A).

Fig. EV5A. The dendritic protrusions from DTC are significantly reduced in *glp-1* mutants. Unpaired *t*-test. Error bars: SD.

Taken together, the reduction of well-accepted cell adhesion markers (HMR-1/E-cadherin & HMP-2/ β -catenin) (Fig. 3B and G) and the change in DTC morphology (Fig. 3B, Fig. EV5A, and Supplementary Movie 1 and 2) are enough to prove a decrease in cell adhesions upon the removal of GSCs.

As for the cytoskeleton in DTC, it is essential to DTC shape and is controlled by germ cells (Agarwal, Shemesh et al., 2022). With such a dramatic change in DTC morphology upon the loss of germ cells, the cytoskeleton in DTC must be accordingly altered, especially in those dendritic structures, as reported in other cell types (Fletcher & Mullins, 2010, Murrell, Oakes et al., 2015). Unlike the change in DTC morphology, which shows a reduction of adhesions with the loss of dendritic protrusions, the detailed change in the cytoskeleton is without the scope of this study. Therefore, we did not further examine the changes in the cytoskeleton. If Reviewer #1 considers it essential, we will clarify this issue further.

Most importantly, the mechanistic link between adhesion alterations and beta-catenin signaling has not been strengthened,

-- Regarding this issue, Reviewer #1's prior comments suggested, 'they need to demonstrate actual increased nuclear localisation of β -catenin in their mutants'. Other two reviewers also raised similar concerns and suggestions.

Following the reviewers' suggestions, we have examined the nuclear localisation of HMP-2/ β -catenin in *glp-1* mutants (Supplementary Fig. 6a) and found an increased nuclear localisation upon germline removal.

Panel a from Supplementary Fig. 6

The nuclear localization of mGreenLantern (mGL)-tagged HMP-2 (arrows) is increased in the DTC of *glp-1* mutants. Scale bar: 10 μ m. Unpaired *t*-test.

This critical new result strengthened the mechanistic link between adhesion alterations and β -catenin, just as Reviewer #1 commented.

on the contrary the new data showing lack of effect *hmr-1*/E-cadherin siRNA on longevity makes the proposed mechanism unlikely

-- There appears to be a misunderstanding of our data.

Instead of a lack of longevity effect, Fig. 3c-f and Supplementary Fig. 5d show that the DTC-specific RNAi against *hmr-1* promotes longevity, including increasing the lifespan of WT worms (Fig. 3c and e, Supplementary Fig. 5d) and upregulating the expression of known gonadal longevity genes through *elt-3* and *pqm-1* (Fig. 3d and f).

Panel c-f from Fig. 3

(c) DTC-specific RNAi against *hmr-1* extends lifespan. Mantel-Cox test.

(d) Inhibiting *hmr-1* in DTC upregulates the expression of indicated genes in WT worms but not in *glp-1* mutants. Error bars: SEM.

(e and f) *elt-3* or *pqm-1* in DTC is required for the extended lifespan (e) and increased expression of *daf-36* and *daf-16* targets (f) induced by DTC-specific RNAi against *hmr-1*.

Panel d from Supplementary Fig. 5

(d) DTC-specific RNAi against *hmr-1* extends lifespan of WT worms but not *glp-1* mutants.

Reviewer #1 could also be referring to the old data in Fig. 3d and new data in Supplementary Fig. 5d, in which *hmr-1* RNAi in DTC does not upregulate gene expression (Fig. 3d) or extend the lifespan of *glp-1* mutants (Supplementary Fig. 5d).

These results fit our conclusion. From the view of genetic epistasis, these results indicate that *hmr-1* (DTC-GSC adhesion) works in the same pathway as removing germline to promote longevity. From the view of cellular mechanisms, DTC-GSC adhesions are already disrupted in the GSC-less *glp-1* mutants, and a further *hmr-1* RNAi can no longer cause a significant effect.

and the connection between altered adhesion and the observed gene expression changes remains unclear.

-- Fig. 3d, f, and i show that altered adhesion, by DTC-specifically suppressing *hmr-1* in WT worms and *hmp-2* in *glp-1* mutants, changes the expression of downstream gonadal longevity genes.

Panels d, f, and i from Fig. 3

(d) Inhibiting *hmr-1* in DTC upregulates the expression of indicated genes in WT worms but not in *glp-1* mutants. Error bars: SEM.

(f) The expression of indicated genes in WT worms subjected to indicated DTC-specific RNAi treatments. Note that *elt-3* or *pqm-1* in DTC is required for the extended lifespan and increased expression of *daf-36* and *daf-16* targets induced by DTC-specific RNAi against *hmr-1*.

(i) DTC-specific RNAi against *hmp-2* inhibits the increased levels of indicated genes in *glp-1* mutants.

The firefly luciferase gene, *luc2*, serves as the negative control in RNAi assays.

Reviewer #2 (Remarks to the Author):

The revised manuscript by Liu et al. is much improved. Several of the key points raised in the prior reviews have been satisfactorily addressed.

-- We are glad to learn that our revision is appreciated by Reviewer #2 (Prof Greenstein).

A few remaining issues are:

1. The authors are making a major claim that germline removal activates the transcription factors ELT-3 and PQM-1 and this point is emphasised in their Abstract. The data on this front are not conclusive. The authors show that germline removal does not affect the expression of ELT-3 or PQM-1 in the DTC (Extended Data Figure 3). They want to conclude that germline removal affects the activity of these transcription factors, but that is not shown. Thus, it remains a speculation. They do provide evidence of low affinity interactions with HMP-2 when the proteins are over-expressed in HEK293 cells (Extended Data Figure 6); however, the scientific literature is littered with these kinds of co-IP experiments in which it can be shown in vitro that proteins can interact. It remains to be determined whether these proteins interact in vivo and whether the interaction is required for promoter binding or transcriptional activation. I think the authors need more compelling data to conclude that these transcription factors are activated by germline removal. I think the fair statement is that these two proteins are required for lifespan extension after germline removal (though see my next point).

-- We first appreciate that Prof Greenstein agrees that the two GATA TFs are 'required' in DTC for lifespan extension after germline removal. We think that the core of our conclusions is the 'requirement' but not the 'activation' of these two GATA TFs. We aim at the source of gonadal longevity signalling and discover that the loss of DTC-GSC adhesion, releases b-catenin from cell membranes, and, in turn, induces transcriptomic changes in DTC through the two GATA TFs. These are the core of our model. So, whether these two GATA TFs are activated is not the core of our model. Our model stands even if the two TFs were not activated, as long as they are 'required'.

Moreover, please note that our bioinformatics analyses strongly suggest that these two TFs are activated in DTC for the transcriptomic changes upon germline removal, because the upregulated genes in DTC are highly enriched of their binding motif (Fig. 1d and e).

We further found that the induction of *tig-2* in DTC upon germline loss is through these

two TFs (Fig. 4a and b), again strongly implying the activation of the two TFs.

These strong hints from gene expression analyses are consistent with our speculation that these two GATA TFs are activated in DTC when the germline is removed. We are glad to rephrase our model more precisely, as Dr Greenstein suggested.

2. In my prior review, I emphasised the concern that the authors over-relied on RNAi approaches in their analyses. This point has not been adequately addressed. Given that apparent null alleles of *elt-3* and *pqm-1* are apparently viable and fertile, readers need to see lifespan assays of both *pqm-1* and *elt-3* null mutants with and without germline removal.

-- We understand Dr Greenstein's concern about the methodology. However, the null alleles of *elt-3* and *pqm-1* are not applicable in this study. We aim to explore the machinery in DTC which triggers gonadal longevity when the germline is removed. Therefore, all interventions in our study are restricted to DTC. *elt-3* and *pqm-1* are widely expressed and play important roles in various worm tissues. So, null alleles will inevitably affect all tissues, and cannot address our scientific question precisely. Therefore, we did not use these mutants in our revision and prior studies.

Meanwhile, we would like to make a brief defense of the DTC-specific RNAi in our study. Tissue-specific RNAi is a well-established and widely used method to suppress gene expression in a specific worm tissue. We have validated our RNAi efficiency in the revised manuscript, as suggested and acknowledged by Prof Greenstein. Besides, we have cross-examined our DTC-specific RNAi data with other methods, such as the DTC-specific gene expression analyses (Fig. 1, 4), DTC microscopy (Fig. 3, S3, S5, S6, S7), and DTC-specific gene overexpression (Fig. 4, S7). These data obtained by other methods are fully consistent with those obtained by DTC-specific RNAi assays. Therefore, we think the RNAi approaches in our study are appropriate and concrete.

3. In line 288 in their Discussion, the authors evoke the idea of a "checkpoint" mechanism. This is a real over-reach here as they have not shown that the genetic or biochemical

criteria for a checkpoint mechanism have been satisfied (Hartwell and Weinert, 1989; Elledge, 1996).

-- We agree with Prof Greenstein's comment. However, this 'checkpoint' idea is NOT part of our model but is a mere speculation from the model. To make our discussion more precise, we will rephrase the text accordingly, such as '...the machinery identified in this study could trigger signalling to rescue reproduction and enhance the resistance of the somatic tissues...'.
.

REV#3 COMMENTS TO THE AUTHORS:

In the manuscript "The niche of germline stem cells induces longevity signalling upon the loss of germline" The authors addressed most of my concerns and the manuscript is substantially improved. I would change a few additional things with regards to the presentation of your lifespan data to make it as accessible to your audience as possible. Right now the readers really need to dig to find the results. Very few readers will look at supplementary tables so you should display the major take aways on the figure itself. That is the percent lifespan extension in that particular replicate (again you can not combine multiple independent replicates into a single figure as you appear to have done in your supplemental table (and maybe in the figures although I can not tell).

-- In figures, only one biological replicate is shown. We have clarified this issue in the revised figure legends.

One independent experiment should be shown in the figure and additional replicate experiments should be shown with their independent statistics in the table. Therefore only one replicate should indicate which figure it is in and the other ones should have no figure indicated next to them. Also do not put the statistics in a separate place further off to the right just represent it in 5 columns as shown below.

-- Many thanks for appreciating our revision! We have reorganized figure panels and supplementary lifespan tables following your suggestions.

Minor points

1) You have to explain what *luc-2* is in the actual text and figure. Right now readers would have to search to figure out that is your control KD.

-- Thanks for your suggestion! We have denoted the use of *luc2* in the legend of each figure with its data.

2) Include the magnitude of lifespan extension and the p value on the figure itself. If you cant fit it all in the figure can put in figure legend at very least.

-- Thanks for your suggestion! The most critical p values are shown on the figure. The magnitude of the most critical median lifespan change is included in the corresponding figure legends.

Due to the limit of space, we can not fit all these information in figures and figure legends. That is indeed a lot of information, as shown in Supplementary Table 3. Therefore, we include a note in the legend of each figure with lifespan analysis to encourage our readers to read our Supplementary Table 3 for all the details.

3) Your supplementary lifespan tables seem to contain more information but are still incoherent. You need to have the mean lifespan of each independent replicate and the statistics for that independent experiment in addition to the # worms alive and censored. This tables should help the readers not be more confusing.

-- Thank you for your suggestion! We have modified the table as suggested.

References

- Agarwal P, Shemesh T, Zaidel-Bar R (2022) Directed cell invasion and asymmetric adhesion drive tissue elongation and turning in *C. elegans* gonad morphogenesis. *Developmental cell* 57: 2111-2126 e6
- Fletcher DA, Mullins RD (2010) Cell mechanics and the cytoskeleton. *Nature* 463: 485-92
- Li X, Singh N, Miller C, Washington I, Sosseh B, Gordon KL (2022) The *C. elegans* gonadal sheath Sh1 cells extend asymmetrically over a differentiating germ cell population in the proliferative zone. *eLife* 11
- Linden LM, Gordon KL, Pani AM, Payne SG, Garde A, Burkholder D, Chi Q, Goldstein B, Sherwood DR (2017) Identification of regulators of germ stem cell enwrapment by its niche in *C. elegans*. *Developmental biology* 429: 271-284
- Murrell M, Oakes PW, Lenz M, Gardel ML (2015) Forcing cells into shape: the mechanics of actomyosin contractility. *Nature reviews Molecular cell biology* 16: 486-98

Dear Dr Yidong Shen,

Thank you again for the submission of your amended manuscript (EMBOJ-2024-116647) to The EMBO Journal, as well as for your patience with our feedback. We have carefully assessed your manuscript and the point-by-point response provided to the referee concerns that were raised during review at a different journal and decided to ask referee #2 from the previous peer-review to evaluate your revision plan, with respect to technical robustness, conceptual advance and overall suitability of your work for publication in The EMBO Journal. Please note that we have in addition asked an arbitrating advisor to assess the work, and have also received input from this expert, which I enclose below.

As you will see from their comments enclosed below, the referee is in favour of the work stating the interest and value of your results but also points to a persistent important shortcoming which needs to be addressed before he can be supportive of publication at The EMBO Journal.

We have discussed the input carefully and concluded that we can accordingly invite you to revise the study along the lines indicated in your response and are pleased to move forward with this work at The EMBO Journal, pending the following remaining issues are addressed in a re-submitted version.

- Complement the existing RNAi interrogation with genetic null mutants and complementary DTC-specific AID constructs to strengthen the claims made on relevance of *elt-3* and *pqm-1* for the germline ablation-induced longevity phenotype (ref#2, pt.2 re-review (Fig3B,C; adv#2)).

Once we have received the revised version, we will have the work reassessed by the expert.

Thank you again for giving us the chance to consider your manuscript for The EMBO Journal, I look forward to hearing from you and receiving your final revised version of the manuscript.

Best regards,

Daniel Klimmeck

Daniel Klimmeck PhD
Senior Editor
The EMBO Journal.

EMBOJ-2024-116788, Referee #2 additional comments:

'There is one significant point that gives me pause for recommending this for EMBO J, which I consider to be a journal that publishes reliable data. The point of concern is that the authors over-rely on RNAi approaches in their analysis. This is a big worry. The concern is that *pqm-1* and *elt-3* are non-essential genes so it should be a simple matter to determine whether they get the same result with mutants as with RNAi against these genes in the DTC. It is standard practice in genetics for ensuring rigor and reproducibility to analyze multiple alleles of a gene or to utilize orthogonal approaches for gene knockdown. In this instance, the expectation would be that DTC RNAi for these genes would give the same (or similar) result to using null alleles, which are viable and fertile.'

External advisor's comments:

The goal of this study is to reveal the molecular processes involved in triggering the the lifespan extension effect upon loss of the germlin in *C. elegans*.

The authors provide data that insinuate a strong contribution by the Distal Tip Cells (DTCs), which constitute the stem cell niche of germ cells. Based on transcriptome analysis of sorted DTCs from wild-type worms and mutants with genetically ablated germlines they linked lifespan effects to the transcription factors ELT-3 and PQM-1, which may interact with the catenin HMP-2 for their activation. Furthermore, by assessing the loss of E-cadherin-based adhesions of DTCs with the germline, they observe increased TIG-2 expression, which is a ligand of TGF- β signaling. The authors conclude that stronger secretion of TIG-2 is responsible for the overall response in lifespan extension.

Major Concerns:

- The interaction of ELT-3 and PQM-1 with HMP-2 in HEK293T cells is not in the physiological context. It does not provide sufficient proof to state with confidence the HMP-2 - ELT-3 / PQM-1 axis. This is in particular important as the authors try to link HMP-2 directly to ELT-3 and PQM-1 in the DTCs. They show the potential for interaction in an artificial cell system but this is far away from the conditions in the DTCs
- How TIG-2 overexpression leads to increased lifespan effects remain unknown. This may be a future prospect but would be the main conclusion of this study.
- The application of DTC-specific RNAi requires the *rde-1* mutant background. This tissue-specific RNAi approach is well established in *C. elegans* and *rde-1* mutants may not show obvious phenotypes. Yet, in the context of aging and in combination with other genotypes it needs to be excluded that the *rde-1* mutation has an effect.
- The tissue-specific depletion of targets such as ELT-3 and PQM-1 requires another approach such as tissue-specific depletion by the auxin-inducible protein degradation (AID) system. Another approach could be the Cre/loxP system which is being used by many *C. elegans* labs (van Heuvel Lab provides strains and DNA constructs). The DTC-specific RNAi is the sole evidence for the claimed effects and needs to be confirmed.
- The assessment of the DTC-specific RNAi based on the GFP constructs (Fig EV4 A) is insufficient. A positive control is missing, as it is possible that non-DTC-specific depletion of the GFP is not detectable. Highly expressed GFP-based reporters (*sur-5::gfp*) show in general only a modest decrease even if targeted directly. The off-tissue depletion may simply be not detectable falsifying the conclusion that there are no off-tissue effects regarding the DTC-specific RNAi.
- Scoring of *tig-2::mcherry* in DTCs (Figure EV7D) reflects a different strength of its increase in *glp-1* background DTCs than shown by the microscopic pictures – there is a notion of implying stronger effects than there are.
- The lifespan increase of *tig-2* overexpression in the intestine (Figure EV7H) is too minor to insinuate a strong contribution of this effect for the overall lifespan extension.
- Several strong statements and use of the term 'critical' (rather than required or contribute) are being used that are not adequate and reflect an attempt to strengthen the conclusion which are not sufficiently validated by experiments:
 1. *Discussion : HMP-2 subsequently alters the transcriptome in DTC though GATA transcription factors ELT-3 and PQM-1. The TGF- β ligand, TIG-2, is thereby upregulated in and secreted from DTC, activating downstream signalling (e.g., DA signalling) in other somatic tissues to promote longevity.*

This statement is an exaggeration of the insinuated links between HMP-2 and the TFs ELT-3 and PQM-1; only an adequate assessment of transcriptome upon HMP-2 increase in the DTC nucleus would allow such an overly strong conclusion and assessment. Would there still be the same transcriptome changes in *glp-1* if HMP-2 is depleted? This should not be the case, if the link is relevant. These details are

important as aging is a multi-tissue effect with many non-cell autonomous effects involved...

2. *“These results then indicate that hmp-2 is critical in the signalling from the DTC-GSC adhesions to GATA TFs in DTC.”*

HMP-2 (should be the protein here; therefore, caps) plays a role but does not seem to be critical!

3. *“Fig. 4 A TGF- β ligand from DTC is critical to gonadal longevity.”*

Required / or contributes rather than critical-

4. *“As the DTC-GSC adhesions were already disrupted upon germline removal (Fig. 3B), inhibiting hmr-1 in the DTC of glp-1 mutants did not affect lifespan or the expression of daf-36 or daf-16 targets (Fig. 3D, Fig. EV5D), confirming that the DTC-GSC adhesions regulates gonadal longevity.”*

No, these results indicate or support the notion that the adhesions contribute to the longevity effects. Such statements are strong exaggerations and are not appropriate based on the provided data.

Minor:

The Cell-specific RNAi needs to be explained in the .

The measurements of nuclear HMP-2 enrichment (EV6A) is not explained in Material and Methods. Where Z-Stacks used?

Fig 4 F should indicate that it comprises all four graphs.

Instruction for the preparation of your revised manuscript:

2) individual production quality figure files as .eps, .tif, .jpg (one file per figure).

3) a .docx formatted letter INCLUDING the reviewers' reports and your detailed point-by-point response to their comments. As part of the EMBO Press transparent editorial process, the point-by-point response is part of the Review Process File (RPF), which will be published alongside your paper.

4) a complete author checklist, which you can download from our author guidelines ([https://wol-prod-cdn.literatumonline.com/pb-assets/embo-site/Author Checklist%20-%20EMBO%20J-1561436015657.xlsx](https://wol-prod-cdn.literatumonline.com/pb-assets/embo-site/Author%20Checklist%20-%20EMBO%20J-1561436015657.xlsx)). Please insert information in the checklist that is also reflected in the manuscript. The completed author checklist will also be part of the RPF.

6) It is mandatory to include a 'Data Availability' section after the Materials and Methods. Before submitting your revision, primary datasets produced in this study need to be deposited in an appropriate public database, and the accession numbers and database listed under 'Data Availability'. Please remember to provide a reviewer password if the datasets are not yet public (see <https://www.embopress.org/page/journal/14602075/authorguide#datadeposition>).

7) Our journal encourages inclusion of *data citations in the reference list* to directly cite datasets that were re-used and obtained from public databases. Data citations in the article text are distinct from normal bibliographical citations and should directly link to the database records from which the data can be accessed. In the main text, data citations are formatted as follows: "Data ref: Smith et al, 2001" or "Data ref: NCBI Sequence Read Archive PRJNA342805, 2017". In the Reference list, data citations must be labeled with "[DATASET]". A data reference must provide the database name, accession number/identifiers and a resolvable link to the landing page from which the data can be accessed at the end of the reference. Further instructions are available at .

8) At EMBO Press we ask authors to provide source data for the main and EV figures. Our source data coordinator will contact you to discuss which figure panels we would need source data for and will also provide you with helpful tips on how to upload and organize the files.

Numerical data can be provided as individual .xls or .csv files (including a tab describing the data). For 'blots' or microscopy, uncropped images should be submitted (using a zip archive or a single pdf per main figure if multiple images need to be supplied for one panel). Additional information on source data and instruction on how to label the files are available at .

9) We replaced Supplementary Information with Expanded View (EV) Figures and Tables that are collapsible/expandable online (see examples in <https://www.embopress.org/doi/10.15252/embo.201695874>). A maximum of 5 EV Figures can be typeset. EV Figures should be cited as 'Figure EV1, Figure EV2' etc. in the text and their respective legends should be included in the main text after the legends of regular figures.

Please remember: Digital image enhancement is acceptable practice, as long as it accurately represents the original data and conforms to community standards. If a figure has been subjected to significant electronic manipulation, this must be noted in the

figure legend or in the 'Materials and Methods' section. The editors reserve the right to request original versions of figures and the original images that were used to assemble the figure.

11) For data quantification: please specify the name of the statistical test used to generate error bars and P values, the number (n) of independent experiments (specify technical or biological replicates) underlying each data point and the test used to calculate p-values in each figure legend. The figure legends should contain a basic description of n, P and the test applied. Graphs must include a description of the bars and the error bars (s.d., s.e.m.).

Responses to the reviewers:

We appreciate the insightful and constructive comments from our reviewers and are grateful for their time and efforts on our manuscript. Please see our point-to-point responses below. All our responses in this letter and the corresponding changes in our revised manuscript are highlighted in blue.

EMBOJ-2024-116788, Referee #2 additional comments:

'There is one significant point that gives me pause for recommending this for EMBO J, which I consider to be a journal that publishes reliable data. The point of concern is that the authors over-rely on RNAi approaches in their analysis. This is a big worry. The concern is that *pqm-1* and *elt-3* are non-essential genes so it should be a simple matter to determine whether they get the same result with mutants as with RNAi against these genes in the DTC. It is standard practice in genetics for ensuring rigor and reproducibility to analyze multiple alleles of a gene or to utilize orthogonal approaches for gene knockdown. In this instance, the expectation would be that DTC RNAi for these genes would give the same (or similar) result to using null alleles, which are viable and fertile.'

-- Following your suggestion, we examined the lifespan of the *pqm-1(ok485)* and *elt-3(gk121)* null mutants (Fig. EV5A). Consistent with our DTC-specific RNAi assays, mutating *pqm-1* or *elt-3* significantly suppressed the longevity of the germline less *glp-1* mutants. By qPCR, we further observed that mutating *elt-3* or *pqm-1* similarly reduced the induction of *daf-36* and *daf-16* targets in *glp-1* mutants (Fig. EV5B). These new results confirm the critical role of these two GATA TFs in gonadal longevity and well support our findings by DTC-specific RNAi.

Fig. EV5 Liu et al.

EMBOJ-2024-116788, Arbitrating advisor's comments

The goal of this study is to reveal the molecular processes involved in triggering the the lifespan extension effect upon loss of the germline in *C. elegans*. The authors provide data that insinuate a strong contribution by the Distal Tip Cells (DTCs), which constitute the stem cell niche of germ cells. Based on transcriptome analysis of sorted DTCs from wild-type worms and mutants with genetically ablated germlines they linked lifespan effects to the transcription factors ELT-3 and PQM-1, which may interact with the catenin HMP-2 for their activation. Furthermore, by assessing the loss of E-cadherin-based adhesions of DTCs with the germline, they observe increased TIG-2 expression, which is a ligand of TGF- β signaling. The authors conclude that stronger secretion of TIG-2 is responsible for the overall response in lifespan extension.

Major Concerns:

1) The interaction of ELT-3 and PQM-1 with HMP-2 in HEK293T cells is not in the physiological context. It does not provide sufficient proof to state with confidence the HMP-2 - ELT-3 / PQM-1 axis. This is in particular important as the authors try to link HMP-2 directly to ELT-3 and PQM-1 in the DTCs. They show the potential for interaction in an artificial cell system but this is far away from the conditions in the DTCs.

-- We agree with your concern. Yet, the coimmunoprecipitation assay in HEK293T cells still suggests an interaction between HMP-2 with the two GATA TFs. We have revised our manuscript accordingly to clarify this issue (page 10) as following:

'Moreover, GFP::HMP-2 was co-immunoprecipitated with ELT-3::FLAG and PQM-1::FLAG (Fig. EV7B and C), suggesting HMP-2 could interact with the two GATA TFs in DTC.'

Our proposed HMP-2 – GATA TFs axis is a genetic pathway. The detailed mechanism underlying how HMP-2 regulates GATA TFs in DTC will be an interesting topic in future studies. In the revised 'Discussion' section, we clarified this issue, as well (page 15):

'Similar to a previously report (Iyer, Nagarajan et al., 2018), we found that HMP-2/ β -catenin could interact with the two GATA TFs, which drives the transcriptomic changes in DTC (Fig. 1, Fig. EV7), suggesting that HMP-2 might directly regulate the two GATA TFs. The detailed mechanism underlying the HMP-2-mediated regulation on the two GATA TFs upon germline removal will be an interesting issue in future studies.'

2) How TIG-2 overexpression leads to increased lifespan effects remain unknown. This may be a future prospect but would be the main conclusion of this study.

-- We agree that how TIG-2 regulates longevity is an interesting issue to pursue. TIG-2 is secreted from DTC and its downstream effectors are likely in other somatic tissues. Therefore, we think that it is beyond the main scope of this manuscript, which focuses on DTC, the origin of the gonadal longevity.

3) The application of DTC-specific RNAi requires the *rde-1* mutant background. This tissue-specific RNAi approach is well established in *C. elegans* and *rde-1* mutants may not show obvious phenotypes. Yet, in the context of aging and in combination with other genotypes it needs to be excluded that the *rde-1* mutation has an effect.

-- We agree with you that it is necessary to check whether *rde-1* mutation could affect gonadal longevity. We then examined the lifespan and the induction of DA signalling and *daf-16* activity (i.e., the upregulation of *daf-36*, *lip1-4*, *dod-8*, and *sod-3*) between *glp-1* and *rde-1;glp-1* mutants (Fig. EV4D and EV4E)(Antebi, 2013), and did not observe a significant difference.

Consistently, our previous assays using *rde-1;glp-1;cpIs121* mutants showed that this strain exhibits a series of features of gonadal longevity when treated with *luc2* RNAi (a negative control in RNAi assays), including the upregulation of *daf-36*, the increase of *daf-16* targets, and the extended lifespan (Antebi, 2013) (Fig. 2, *rde-1;glp-1;cpIs121* is denoted as *glp-1*). Therefore, we think that *rde-1* mutation should not interfere with gonadal longevity.

Fig. 2 Liu et al.

Moreover, tissue-specific RNAi based on *rde-1* mutation is widely used in the ageing research of other longevity mutants, such as these reports in *daf-2* mutants (Rasulova, Zecic et al., 2021, Son, Seo et al., 2017, Uno, Tani et al., 2021), supporting that this mutation has no obvious effect in the context of ageing.

4) The tissue-specific depletion of targets such as ELT-3 and PQM-1 requires another approach such as tissue-specific depletion by the auxin-inducible protein degradation (AID) system. Another approach could be the Cre/loxP system which is being used by many *C. elegans* labs (van Heuvel Lab provides strains and DNA constructs). The DTC-specific RNAi is the sole evidence for the claimed effects and needs to be confirmed.

-- Thank you for your comment! As also suggested by the other reviewer, we examined the lifespan, and the expression of *daf-36* and *daf-16* targets in *elt-3* and *pqm-1* null mutants (Fig. EV5). Our observation that mutating *elt-3* or *pqm-1* significantly suppressed the gonadal longevity of *glp-1* mutants well supports our findings with DTC-specific RNAi against these two GATA transcription factors.

Fig. EV5 Liu et al.

5) The assessment of the DTC-specific RNAi based on the GFP constructs (Fig EV4 A) in insufficient. A positive control is missing, as it is possible that non-DTC-specific depletion of the GFP is not detectable. Highly expressed GFP-based reporters (*sur-5::gfp*) show in general only a modest decrease even if targeted directly. The off-tissue depletion may simply be not detectable falsifying the conclusion that there are no off-tissue effects regarding the DTC-specific RNAi.

-- Many thanks for reminding us of this critical positive control!

Following your suggestion, we treated wild-type N2 worms expressing *sydEx229[myo-2p::mCherry::unc-54u; sur-5p::gfp]* with *gfp* RNAi (Fig. EV4C). *sydEx229* is the same transgene expressing *sur-5p::gfp* in our assessment of the DTC-specific RNAi (Fig. EV4A and B). As we expected, *gfp* RNAi significantly suppressed the expression of *sur-5p::gfp* in major somatic tissues (e.g., intestine, arrows in Fig. EV4C) but not in neurons, which are resistant to RNAi

treatment (arrowheads in Fig. EV4C). Therefore, non-DTC-specific depletion of the GFP, if any, can be easily detected in our assays.

The transgene we used in our assays is a new one prepared by ourselves. Its expression could be weaker than the reported ones you mentioned. Therefore, it shows a clear decrease upon *gfp* RNAi treatment.

6) Scoring of *tig-2::mcherry* in DTCs (Figure EV7D) reflects a different strength of its increase in *gfp-1* background DTCs than shown by the microscopic pictures - there is a notion of implying stronger effects than there are.

-- We agree with your comment. The quantification of fluorescent intensity is a semi-quantitative method and may not always reflect the exact fold of change, especially when the fluorescent signal is mild (e.g., the case of *tig-2p::tig-2::mCherry*). That is exactly why we presented a representative image side by side.

7) The lifespan increase of *tig-2* overexpression in the intestine (Figure EV7H) is too minor to insinuate a strong contribution of this effect for the overall lifespan extension.

-- We agree that *tig-2* overexpression in the intestine did not extend lifespan as strong as its overexpression in DTC (Fig. 4F and EV7H). But it still extends the median lifespan by ~10% and maximum lifespan by ~15% with statistical significance (Table S3). Therefore, this assay supports our conclusion that TIG-2 functions as a secreted protein in gonadal longevity.

Because intestine is a tissue with little *tig-2* expression (Wang, Jiang et al., 2022), the artificial expression of *tig-2* in it may cause some unknown side effects, causing the weaker extension of lifespan.

8) Several strong statements and use of the term 'critical' (rather than required or contribute) are being used that are not adequate and reflect an attempt to strengthen the conclusion which are not sufficiently validated by experiments:

-- Thank you for the critics! We have revised our manuscript as suggested. Please see our detailed response below:

1. Discussion: HMP-2 subsequently alters the transcriptome in DTC through GATA transcription factors ELT-3 and PQM-1. The TGF- β ligand, TIG-2, is thereby upregulated in and secreted from DTC, activating downstream signalling (e.g., DA signalling) in other somatic tissues to promote longevity.

This statement is an exaggeration of the insinuated links between HMP-2 and the TFs ELT-3 and PQM-1; only an adequate assessment of transcriptome upon HMP-2 increase in the DTC nucleus would allow such an overly strong conclusion and assessment. Would there still be the same transcriptome changes in *glp-1* if HMP-2 is depleted? This should not be the case, if the link is relevant. These details are important as aging is a multi-tissue effect with many non-cell autonomous effects involved...

-- As commented, we have rephrased our statement as '*The loss of GSCs disrupts the cell adhesions between GSC and DTC, leading to the release of HMP-2/ β -catenin in DTC from the*

cell membrane and the transcriptomic changes in DTC through GATA transcription factors ELT-3 and PQM-1' (page 12).

2. "These results then indicate that hmp-2 is critical in the signalling from the DTC-GSC adhesions to GATA TFs in DTC." HMP-2 (should be the protein here; therefore, caps) plays a role but does not seem to be critical!

-- The statement has been rephrased as '*These results then indicate that HMP-2 functions in the signalling from the DTC-GSC adhesions to GATA TFs in DTC*' (page 10).

3. "Fig. 4 A TGF- β ligand from DTC is critical to gonadal longevity." Required / or contributes rather than critical-

-- The title of Fig. 4 has been rephrased as '*A TGF- β ligand from DTC is required for gonadal longevity*' (page 40).

4. "As the DTC-GSC adhesions were already disrupted upon germline removal (Fig. 3B), inhibiting hmr-1 in the DTC of glp-1 mutants did not affect lifespan or the expression of daf-36 or daf-16 targets (Fig. 3D, Fig. EV5D), confirming that the DTC-GSC adhesions regulates gonadal longevity."

None of these results indicate or support the notion that the adhesions contribute to the longevity effects. Such statements are strong exaggerations and are not appropriate based on the provided data.

-- We are sorry for our confusing phrasing. The direct evidence showing that the DTC-GSC adhesions regulate longevity is the data of *hmr-1* RNAi in WT worms. In these data, which are mentioned in the same paragraph, DTC-specific RNAi against *hmr-1* disrupts DTC-GSC adhesions, increases lifespan, and upregulates *daf-36* and *daf-16* targets (Fig. 3C, 3D, and EV6C).

hmr-1 RNAi in *glp-1* mutants, which no longer have DTC-GSC adhesions, is to check whether *hmr-1* functions through the DTC-GSC adhesions. If DTC-specific RNAi against *hmr-1* could

extend the lifespan of *glp-1* or increase the expression of *daf-36* and *daf-16* targets, it should have controlled gonadal longevity independent of the DTC-GSC adhesions. Because DTC-specific RNAi against *hmr-1* in *glp-1* mutants had no such phenotypes (Fig. 3C, 3D, and EV6D), we conclude that *hmr-1* regulates gonadal longevity through the DTC-GSC adhesions and these adhesions control gonadal longevity.

We have revised the paragraph in page 8 as shown below:

'Disrupting the DTC-GSC adhesions via DTC-specific hmr-1 RNAi in WT worms from L3 extended lifespan (Fig. 3C, Fig. EV6C) and upregulated daf-36 and the target genes of daf-16 (Fig. 3D), supporting that the DTC-GSC adhesions regulate gonadal longevity. The DTC-GSC adhesions were already disrupted upon germline removal (Fig. 3B). We then knocked down hmr-1 in the DTC of glp-1 mutants to confirm whether it controls gonadal longevity through the DTC-GSC adhesions. If so, the DTC-specific RNAi against hmr-1 should have no corresponding phenotypes. Indeed, inhibiting hmr-1 in the DTC of glp-1 mutants did not affect lifespan or the expression of daf-36 or daf-16 targets (Fig. 3D, Fig. EV6D).'

Minor:

1) The Cell-specific RNAi needs to be explained.

-- As commented, the strategy of DTC-specific RNAi is explained in the legend of Fig. EV4A (page 46) (Linden, Gordon et al., 2017), as shown below:

'The strain for DTC-specific RNAi is a rde-1 mutant with rde-1 rescued in DTC via a single copy transgene, as reported by Sherwood lab.'

2) The measurements of nuclear HMP-2 enrichment (EV6A) is not explained in Material and Methods.

-- As commented, corresponding description is included at the end of the 'Microscopy' section in Material and Methods, as shown below:

'For the nuclear HMP-2 enrichment in DTC, the DTC nucleus was selected and the fluorescence intensity of mGL::HMP-2 was measured in the selected area.'

3) Fig 4 F should indicate that it comprises all four graphs.

-- We are sorry for the confusing organization of Fig. 4F. The four graphs of Fig. 4F have been reorganised in a row for a clearer presentation, as shown below:

Fig. 4 Liu et al.

References

- Antebi A (2013) Regulation of longevity by the reproductive system. *Experimental gerontology* 48: 596-602
- Iyer LM, Nagarajan S, Woelfer M, Schoger E, Khadjeh S, Zafiriou MP, Kari V, Herting J, Pang ST, Weber T, Rathjens FS, Fischer TH, Toischer K, Hasenfuss G, Noack C, Johnsen SA, Zelarayan LC (2018) A context-specific cardiac beta-catenin and GATA4 interaction influences TCF7L2 occupancy and remodels chromatin driving disease progression in the adult heart. *Nucleic acids research* 46: 2850-2867
- Linden LM, Gordon KL, Pani AM, Payne SG, Garde A, Burkholder D, Chi Q, Goldstein B, Sherwood DR (2017) Identification of regulators of germ stem cell enwrapment by its niche in *C. elegans*. *Developmental biology* 429: 271-284
- Rasulova M, Zecic A, Monje Moreno JM, Vandemeulebroucke L, Dhondt I, Braeckman BP (2021) Elevated Trehalose Levels in *C. elegans* daf-2 Mutants Increase Stress Resistance, Not Lifespan. *Metabolites* 11
- Son HG, Seo M, Ham S, Hwang W, Lee D, An SW, Artan M, Seo K, Kaletsky R, Arey RN, Ryu Y, Ha CM, Kim YK, Murphy CT, Roh TY, Nam HG, Lee SV (2017) RNA surveillance via nonsense-mediated mRNA decay is crucial for longevity in daf-2/insulin/IGF-1 mutant *C. elegans*. *Nature communications* 8: 14749
- Uno M, Tani Y, Nono M, Okabe E, Kishimoto S, Takahashi C, Abe R, Kurihara T, Nishida E (2021) Neuronal DAF-16-to-intestinal DAF-16 communication underlies organismal lifespan extension in *C. elegans*. *iScience* 24: 102706
- Wang X, Jiang Q, Song Y, He Z, Zhang H, Song M, Zhang X, Dai Y, Karalay O, Dieterich C, Antebi A, Wu L, Han JJ, Shen Y (2022) Ageing induces tissue-specific transcriptomic changes in *Caenorhabditis elegans*. *The EMBO journal* 41: e109633

Dear Dr Yidong Shen,

Thank you again for the submission of your amended manuscript (EMBOJ-2024-116788-T) to The EMBO Journal. We have carefully assessed your manuscript and the point-by-point response provided to the referee concerns that were raised during peer review at a different venue, as well as your response to the additional arbitrating advisor. We have received a re-report by referee #2 who states that the key concern on technical robustness of your findings (ref#2; see also arbitrating advisor pts. 3,4) was adequately addressed and s/he is now in favour of publication of the work. As mentioned before, we consider the issues raised by the arbitrator regarding physiological relevance and mechanistic depth of your findings (arbitrating advisor pts. 1,2,7) as well taken as such but overall not required for completion of the current study. Also, we noted that you toned down claims according to this expert's suggestion (arbitrating advisor pt 8).

We are thus pleased to inform you that the study is now provisionally accepted for publication at the EMBO Journal, and that we are prepared to swiftly move forward towards acceptance of this work.

We still need you to take care of a number of minor issues related to formatting and data annotation, as detailed below, as well as full provision of Source Data for the study.

Please let us know any time if there are questions related and submit a revised version of the manuscript using the link enclosed below.

As you might have seen on our web page, every paper at the EMBO Journal now includes a 'Synopsis', displayed on the html and freely accessible to all readers. The synopsis includes a 'model' figure as well as 2-5 one-short-sentence bullet points that summarize the article. I would appreciate if you could provide this figure and the bullet points.

Thank you again for giving us the chance to consider your manuscript for The EMBO Journal, I look forward to hearing from you and receiving your final revised version of the manuscript.

Best regards,

Daniel Klimmeck

Daniel Klimmeck PhD
Senior Editor
The EMBO Journal.

>> Please add up to five keywords to your study.

>> Author Contributions: Please remove the author contributions information from the manuscript text. Note that CRediT has replaced the traditional author contributions section as of now because it offers a systematic machine-readable author contributions format that allows for more effective research assessment. and use the free text boxes beneath each contributing author's name to add specific details on the author's contribution.

More information is available in our guide to authors.
<https://www.embopress.org/page/journal/14602075/authorguide>

>> Adjust the title of the 'Declaration of Interests' section to 'Disclosure and Competing Interests Statement' and move after Acknowledgements.

>> Figures: remove the figures from the manuscript file.

>> Figure callouts: Fig. EV9 needs to be called out.

>> Change the current 'Main' paragraph to 'Introduction'.

>> Limit the abstract to maximally 175 words and remove literature references.

>> There are currently nine EV figures, but only accommodate up to five can be accommodated. Please present the rest in one .pdf appendix file called 'Appendix' with ToC (and page numbers) on its first page; adjust the figure nomenclature and callouts to 'Appendix Figure S1, 2' etc.

>> Dataset EV Legends: Supplementary/Appendix Tables 1-4 are datasets so they need to be updated as Dataset EV1-EV4 (source file names, legends, callouts in the manuscript need to be updated); the legends need to be removed from the manuscript file; Suppl. Table 3 should be made source data.

>> Movies: nomenclature, legends and callouts need to be corrected to Movie EV1 and Movie EV2; movie legends need to be removed from the manuscript file and each should be provided as a readme.txt file; each movie should be zipped up with its corresponding legend and uploaded as folder per movie.

>> Supplementary Material section on p.42 needs to be removed; the title for the EV figure should be Expanded View Figures.

>> Please provide source data for the study as to the separate request e-mail and source data list by my colleague Hannah Sonntag.

>> Reference your NatComms 2019 study in the Methods part.

>> References: adjust reference format to EMBO Journal format, 10 authors plus 'et al.' .

>> Data availability section: remove the referee token and make sure privacy is released from the online dataset.

>> Consider additional changes and comments from our production team as indicated below:

- Data Availability Section:

1. Please note that the accession ID for the GEO database is not provided separately but with the specific URL, in the data availability statement.

2. Please note that reviewer access codes for GEO dataset is provided in the manuscript.

- Figure legends:

1. Please note that the individual figure legends for figures EV 3a-b is not provided in the legends. This needs to be rectified.

2. Please note that the exact p values are not provided in the legends of figures 1c; 2b-d; 3b, d-i; 4a, c-f; EV 1d, g; EV 5a-b; EV 6a, d-e; EV 7a.

3. Please indicate the statistical test used for data analysis in the legends of figures 1d; EV 2; EV 8e.

4. Although 'n' is provided, please describe the nature of entity for 'n' in the legend of figure EV 8a.

5. Please note that the error bars are not defined in the legends of figures 3b, f-g, i; 4b; EV 1c-d; EV 3b; EV 4b, e; EV 5b; EV 6a; EV 7a; EV 8a, c-e.

Referee #1:

The authors have satisfactorily addressed the key points from the prior critiques. I think the finding that a DTC TGF-beta-related ligand, TIG-2, mediates lifespan extension by germline "ablation" will engender much interest in the field.

David Greenstein

The authors addressed the minor editorial issues.

Dear Dr Shen,

Thank you for submitting the revised version of your manuscript. I have now evaluated your amended manuscript and concluded that the remaining minor concerns have been sufficiently addressed.

I am thus pleased to inform you that your manuscript has been accepted for publication in the EMBO Journal.

On a different note, I would like to alert you that EMBO Press offers a format for a video-synopsis of work published with us, which essentially is a short, author-generated film explaining the core findings in hand drawings, and, as we believe, can be very useful to increase visibility of the work. Please see the following link for representative examples and their integration into the article web page:

<https://www.embopress.org/doi/full/10.15252/emj.2019103932>

Best regards,

Daniel Klimmeck

Daniel Klimmeck, PhD
Senior Editor
The EMBO Journal
EMBO
Postfach 1022-40
Meyerhofstrasse 1
D-69117 Heidelberg
contact@embojournal.org
Submit at: <http://emboj.msubmit.net>
